# Embryos and embryonic stem cells from the white rhinoceros

Thomas B. Hildebrandt[1,2], Robert Hermes[1], Silvia Colleoni[3], Sebastian Diecke[4,5], Susanne Holtze[1], Marilyn B. Renfree [6], Jan Stejskal[7], Katsuhiko Hayashi[8], Micha Drukker[9], Pasqualino Loi[10], Frank Göritz[1], Giovanna Lazzari[3,11] & Cesare Galli [3,11]

The northern white rhinoceros (NWR, *Ceratotherium simum cottoni*) is the most endangered mammal in the world with only two females surviving. Here we adapt existing assisted reproduction techniques (ART) to fertilize Southern White Rhinoceros (SWR) oocytes with NWR spermatozoa. We show that rhinoceros oocytes can be repeatedly recovered from live SWR females by transrectal ovum pick-up, matured, fertilized by intracytoplasmic sperm injection and developed to the blastocyst stage in vitro. Next, we generate hybrid rhinoceros embryos in vitro using gametes of NWR and SWR. We also establish embryonic stem cell lines from the SWR blastocysts. Blastocysts are cryopreserved for later embryo transfer. Our results indicate that ART could be a viable strategy to rescue genes from the iconic, almost extinct, northern white rhinoceros and may also have broader impact if applied with similar success to other endangered large mammalian species.

[1] Leibniz Institute for Zoo and Wildlife Research, D-10315 Berlin, Germany. [2] Freie Universität Berlin, D-14195 Berlin, Germany. [3] Avantea, Laboratory of Reproductive Technologies, 26100 Cremona, Italy. [4] Max Delbrück Center, Berlin, Germany. [5] Berlin Institute of Health, Berlin, Germany. [6] School of BioSciences, The University of Melbourne, Melbourne, VIC 3010, Australia. [7] ZOO Dvůr Králové, Štefánikova 1029, 544 01 Dvůr Králové nad Labem, Czech Republic. [8] Department of Stem Cell Biology and Medicine, Graduate School of Medical Sciences, Kyushu University, Fukuoka 812-0054, Japan. [9] Institute of Stem Cell Research and the Induced Pluripotent Stem Cell Core Facility, Helmholtz Center Munich, 85764 Neuherberg, Germany. [10] Faculty of Veterinary Medicine, University of Teramo, Teramo, Italy. [11] Fondazione Avantea, 26100 Cremona, Italy. These authors contributed equally: Thomas B. Hildebrandt, Robert Hermes. These authors jointly supervised this work: Giovanna Lazzari, Cesare Galli. Correspondence and requests for materials should be addressed to T.B.H. (email: HILDEBRAND@izw-berlin.de) or to C.G. (email: cesaregalli@avantea.it)

Conventional approaches toward establishing a self-sustaining NWR population over the last two decades have been repeatedly unsuccessful. To date only one study related to ovum pick up and embryos production in rhinoceros has been reported[1]. There has been no report of the production of rhinoceros embryo from fertilization to the pre-implantation stage[2,3]. Our results suggest that ART might offer an option for rescuing genes from the NWR, an essential first step in saving this nearly extinct rhinoceros sub-species. Beyond applying this classical ART approach, it would be critical to generate artificial gametes. ES cells are the best source for producing primordial germ cells and also represent the "gold standard" for assessing artificial gametes derived from induced pluripotent stem cells (iPSCs)[4]. A holistic strategy combining all three steps would help to maximize genetic diversity in the NWR by providing cryo-banked spermatozoa and somatic cells of several NWR individuals[5]. The rhinoceros and horse share a common ancestor[6], therefore, assisted reproduction techniques (ART) developed in equines[7,8] can potentially be translated to rhinoceros species, including the Southern White Rhinoceros (SWR, *Ceratotherium simum simum*), a closely related sub-species now not at risk of extinction[9]. Using this technology, the NWR faces some unique challenges; there are only two surviving NWRs and both are infertile. They are large, somewhat intractable animals, bigger than horse and difficult to manipulate, and ART has never been completely successful in this species[1]. Reproduction through assisted medical procedures requires safely harvesting oocytes and spermatozoa, achieving in vitro maturation of oocytes, in vitro fertilization, stimulating the resulting zygote to grow to the blastocyst stage, and safely and efficiently transferring the blastocyst to the uterus of a synchronous surrogate mother. Surplus embryos can be cryopreserved for later transfer or processed as sources of stem cell. Here, we show that rhinoceros oocytes can be repeatedly recovered from live SWR females by transrectal ovum pick-up (OPU), matured in vitro, fertilized by intracytoplasmic sperm injection (ICSI) and, for the first time, develop to the blastocyst stage in vitro. Due to a lack of sufficient numbers of NWR oocytes (total 4, harvested post mortem), we injected SWR oocytes with NWR spermatozoa. We also established cell lines from the SWR blastocysts that had typical features of embryonic stem (ES) cells, confirming the potential viability of these rhinoceros embryos. Pure-bred SWR and hybrid fSWR × mNWR blastocysts were cryopreserved for later embryo transfer. Our results indicate that ART could be a viable strategy to rescue genes from the iconic, almost extinct, Northern White rhinoceros and may also have broader impact if applied with similar success to other endangered large mammalian species.

## Results

### Ovarian stimulation and transrectal oocyte recovery.
To harvest oocytes from rhinoceros species, we developed protocols for ovarian stimulation (Supplementary Figure 1) to increase the number of available follicles, as well as safe procedures for general anesthesia and for transrectal oocyte recovery[1]. Based on our preliminary data and the large body size of WR females (fWR), a rhinoceros-specific oocyte collection instrument (patent pending) was developed in order to reach the ovaries through the rectum. This also required a long double-lumen needle and tubing connected to an aspiration pump to recover the fluid aspirated from the follicles (Fig. 1). The donor fSWRs are distributed across different European zoos, and in each case the collection of oocytes was performed on-site (with the inherent difficulties of transporting all required materials). All follicles visible in the ovaries at a size of 1 cm or larger were aspirated using an ultrasound-guided double lumen 16 Gauge needle, in order to repeatedly flush them

to dislodge the oocyte, as is the procedure in the horse[8] (Fig. 1). A total of 18 OPU procedures were performed (Table 1) including a post mortem collection from a deceased fNWR for natural causes related to age (Supplementary figure 2). We recovered 83 oocytes (26.4%) from the 314 follicles that were punctured (Table 1). Oocytes were incubated in holding media for 24–36 h at 22 °C during transportation to the laboratory in Italy to Cremona by express courier for subsequent in vitro maturation, fertilization, and embryo culture[7,10].

### Sperm selection for ICSI.
Obtaining competent spermatozoa is crucial in the successful development of embryos, in terms of achieving normal fertilization, activating the oocyte, and its healthy development to the blastocyst stage. Good quality semen, fresh or frozen, is available from mSWR[1,2,11–13]. For mNWR, the situation is more critical. The amount of semen is limited, is of poor quality, and has been obtained from only three bulls. This made ICSI, a routine procedure in horses, the only option that would likely lead to success[14]. We obtained samples of the sperm available from the males and tested their quality by measuring their ability to sustain pronuclei formation and initiate early cleavage in pig oocytes following ICSI. We assessed the frozen semen of three mNWR and five mSWR in this way. After thawing, the mNWR samples had poor or no motility and led to low rates of fertilization (as determined by pronuclei formation) and cleavage, in contrast to the mSWR spermatozoa, although they also exhibited variable motility after thawing (Supplementary Table 1). The results were, however, improved through the introduction of an electrical activation protocol soon after sperm injection (two DC pulses of 1.5 kV/cm for 30 μm seconds), a technique used for cell fusion and activation of oocytes after nuclear transfer in livestock species[15,16] and in human-assisted reproduction clinics defined as assisted oocyte activation (AOA), used in cases of failure to form pronuclei after ICSI and or to undergo cleavage[17–19]. This dramatically increased the formation rate of pronuclei (from 30 to 90% fertilization rate from one mNWR) and cleavage (from 37 to 67%) in the xenogenic pig oocyte test for two out of the three bull's samples tested. Following these results, we routinely used this protocol when mNWR sperm was used for ICSI. We also excluded the low-performing mNWR3 sperm sample from our further work as it would have negatively affected embryo development.

### In vitro oocyte maturation and ICSI and embryo development.
Upon reaching the laboratory, oocytes were transferred to maturation media and incubated at 37.5 °C in 5% $CO_2$ for 36–44 h until a polar body (PB) was detected. Thirty-two oocytes (38.6%) reached the metaphase II stage suitable for ICSI. Overall, of 19 oocytes injected with mSWR1 sperm, six underwent cleavage, and three developed to blastocysts. Of 13 injected with mNWR, six cleaved and four developed into blastocysts (Table 1). The amount of time required for maturation was from 36 to 44 h and the time required to reach morula compaction and blastulation was 7–9 and 9–12 days, respectively. These periods were relatively long compared to livestock species that reach the same stages at day 5 and 8 after fertilization[20]. Four embryos were used for stem cell derivation (two SWR × SWR, 051A and 051B and two SWR × NWR, 240A and 240B). Three embryos (one SWR × SWR, two SWR × NWR) were cryopreserved for later transfer to recipient females, since the embryo transfer procedure has yet to be developed and validated in rhinoceroses. A high proportion of embryos that started cleavage underwent compaction and blastulation (Fig. 2) and were of high quality. Most of them had a clearly detectable inner cell mass (ICM).

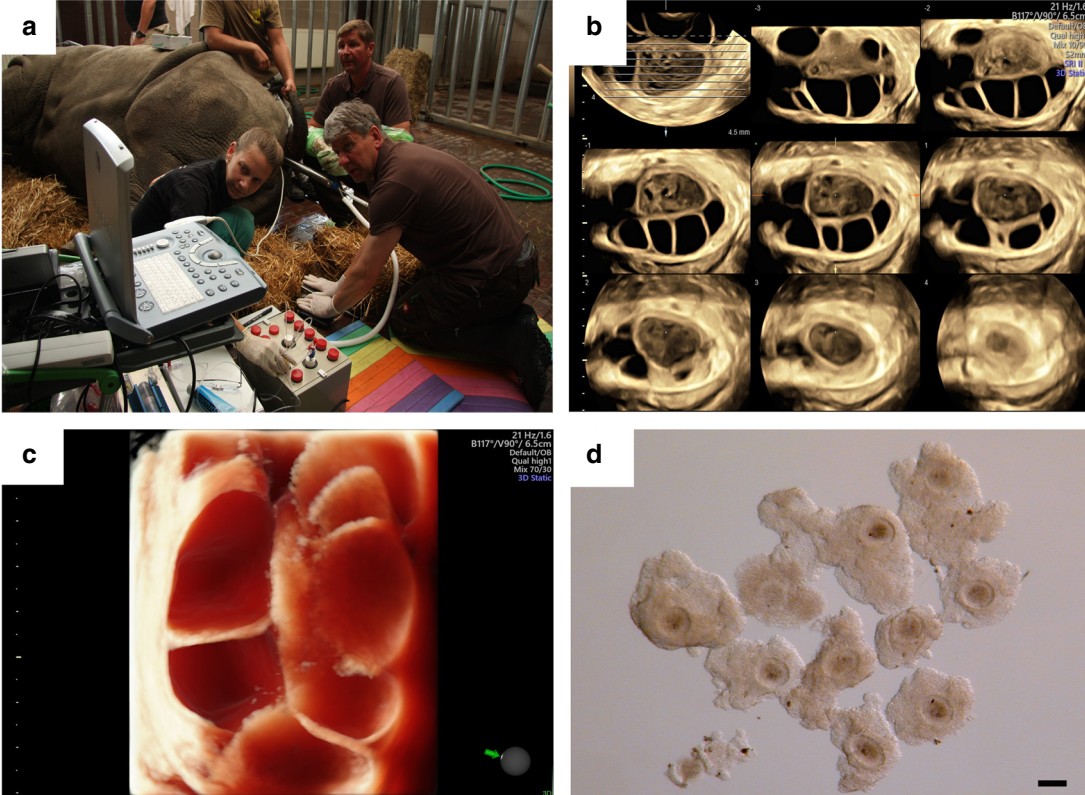

**Fig. 1** Operational setting of the transrectal oocyte retrieval. **a** An anaesthetized white rhinoceros positioned in sternal recumbency during oocyte collection. One operator guides the OPU instrument inside the rectum directly to the stimulated ovary. The second operator injects the needle into the follicles under ultrasound guidance and performs the aspiration and repeatedly flushing the follicle up to ten times. The third operator replaces the full sterile 50-ml falcon tubes with an empty one. The 3D tomographic ultrasound image (**b**) generated with the portable 3D/4D ultrasound unit Voluson I (GE Healthcare, Solingen, Germany) shows different slices of the ovarian parenchyma which contain one central echogenic corpus luteum (white asterisk) and many anechoic follicles ranging from 4 to 20 mm in diameter. The 3D render mode ultrasound image (**c**) shows a virtual cut through the stimulated rhinoceros ovary. In this section eight different follicles are visible. **d** Ten rhinoceros oocytes after transrectal oocyte retrieval. Most of the oocytes collected had an intact cumulus oocyte complex besides the fact that the double-lumen aspiration needle used has a length of 950 mm and the vacuum applied was 100 mmHg. Bar 100 μm

**Table 1 Ovum pick-up and embryo development following in vitro maturation and ICSI**

| Date | Donor | No. follicles | No. oocytes | Recovery rate% | Injected (Metaph. II) (%) | Cleaved (%) | Blastocyst | Bull |
|---|---|---|---|---|---|---|---|---|
| 29/07/15 | fNWR[a] | 5 | 4 | 80.00 | 2 | 0 | 0 | mSWR1 |
| 13/10/15 | fSWR1 | 11 | 4 | 36.36 | 0 | 0 | 0 | mSWR1 |
| 07/01/16 | fSWR2 | 18 | 2 | 11.11 | 1 | 0 | 0 | mSWR1 |
| 23/02/16 | fSWR3[1] | 19 | 3 | 15.79 | 0 | 0 | 0 | mSWR1 |
| 09/05/16 | fSWR4 | 18 | 5 | 27.78 | 1 | 1 | 0 | mSWR1 |
| 11/05/16 | fSWR5[1] | 33 | 3 | 9.09 | 1 | 0 | 0 | mSWR1 |
| 27/09/16 | fSWR6 | 14 | 3 | 21.43 | 2 | 1 | 0 | mSWR1 |
| 27/09/16 | fSWR7 | 15 | 6 | 40.00 | 4 | 0 | 0 | mSWR1 |
| 18/10/16 | fSWR3[2] | 20 | 4 | 20.00 | 2 | 0 | 0 | mSWR1 |
| 07/11/16 | fSWR8[1] | 10 | 5 | 50.00 | 2 | 0 | 0 | mSWR1 |
| 20/02/17 | fSWR5[2] | 37 | 7 | 18.92 | 2 | 2 | 2 | mSWR1 |
| 27/03/17 | fSWR4 | 22 | 6 | 27.27 | 2 | 2 | 1 | mSWR1 |
| 09/04/17 | fSWR8[2] | 26 | 3 | 11.54 | 2 | 2 | 1 | mNWR2 |
| 12/06/17 | fSWR5[3] | 20 | 8 | 40.00 | 1 | 0 | 0 | mNWR1 |
| 17/07/17 | fSWR9 | 15 | 8 | 53.33 | 5 | 0 | 0 | mNWR2 |
| 18/07/17 | fSWR10 | 6 | 1 | 16.67 | 1 | 0 | 0 | mNWR1 |
| 18/07/17 | fSWR11 | 1 | 0 | 0.00 | 0 | 0 | 0 | / |
| 28/08/17 | fSWR5[4] | 24 | 11 | 45.83 | 4 | 4 | 3 | mNWR2 |
| | Totals | 314 | 83 | 26.43 | 32 | 12 | 7 | |
| | | | | | 38.55 | 14.46 | 8.43 | |

Superscripts indicate the number of repeated OPU on the same donor
[a] Oocytes were harvested from the ovaries after the natural death of the animal being the only NWR female used

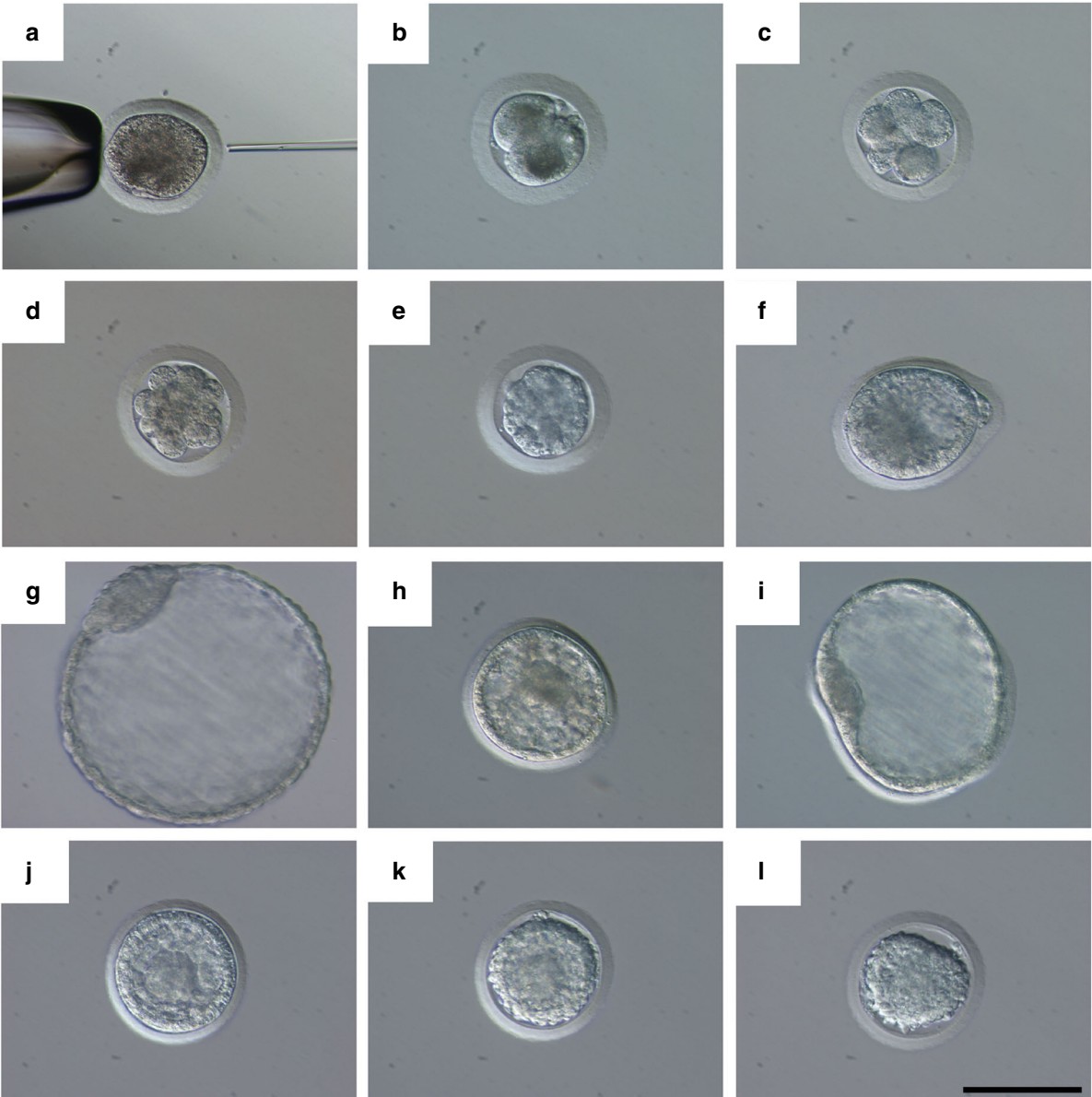

**Fig. 2** Embryo development from ICSI to blastocyst stage. **a** ICSI was performed on MII oocytes with a Piezo-driven needle, the first polar is visible at 6 o'clock, a sperm is visible inside the ICSI pipette ready for injection. A two-cell embryo 48 h after ICSI (**b**). Cleaved embryos developed to the eight cell stage on average 5–6 days after injection (**c**) and were at pre-compaction stage on day 5–7 (**d**). Compacted morulae were visible on average at day 7–9 (**e**). The first blastocyst produced was a pure Southern White Embryo (fSWR5 × mSWR1, day 11 after ICSI) (**f**). The zona was removed by pronase digestion and embryo was left to expand for 2 days. From the isolated ICM of this embryo (**g**) and its twin obtained from the same group of oocytes (fSWR5 × mSWR1) two ES cell lines were derived. Subsequently Hybrid Southern White × Northern White blastocysts were obtained (**h**) (fSWR5 × mNWR1, day 9 after ICSI). One of these hybrid blastocysts (fSWR5 × mNWR2) naturally hatched from the zona at day 10 (**i**) and was used for ES cell derivation. One Southern blastocyst (fSWR4 × mSWR1) and two hybrid blastocysts (fSWR5 × mNWR2 and fSWR8 × mNWR2 (**j**)) were frozen with a standard slow freezing protocol after equilibration in 5% and subsequently in 10% of glycerol freezing media (fSWR8 × mNWR2 (**k**, **l**)). Scale bar corresponds to 100 μm

**ES cell line derivation and differentiation**. To indirectly assess the quality of such embryos and with the prospect of deriving germ cells from stem cells[4,5], we used four embryos for stem cell derivation following protocols previously established in cattle[21]. ES cell lines provide by comparison the best assessment of the quality of existing rhinoceros iPS cell lines[22]. The two SWR × SWR embryos plated at day 14 and day 15, exhibited clearly visible, large ICMs while the two SWR × NWR embryos plated at day 12 and day 13, had considerably smaller ICMs. All the embryos were micromanipulated with fine needles. The mechanically isolated ICMs and trophoblasts were plated separately on inactivated mouse embryonic fibroblasts (MEFs) and cultured in medium DMEM-F12 supplemented with serum replacement and bFGF according to protocols commonly in use for human ES cell cultures (Fig. 3). When a sufficient number of undifferentiated cells (characterized by high nucleus/cytoplasm ratios and large nucleoli in the nucleus, Supplementary Figure 3) were available, we cryopreserved a proportion of them to secure the lines. We expanded some for stem cell marker analysis, parentage, and sex determination (Supplementary Table 5) and to test their differentiation potential in vitro (Fig. 3). We also confirmed viability after repeated rounds of cryopreservation and thawing. From each of the two SWR × SWR female embryos, we established a cell line based on an expansion of the isolated ICM

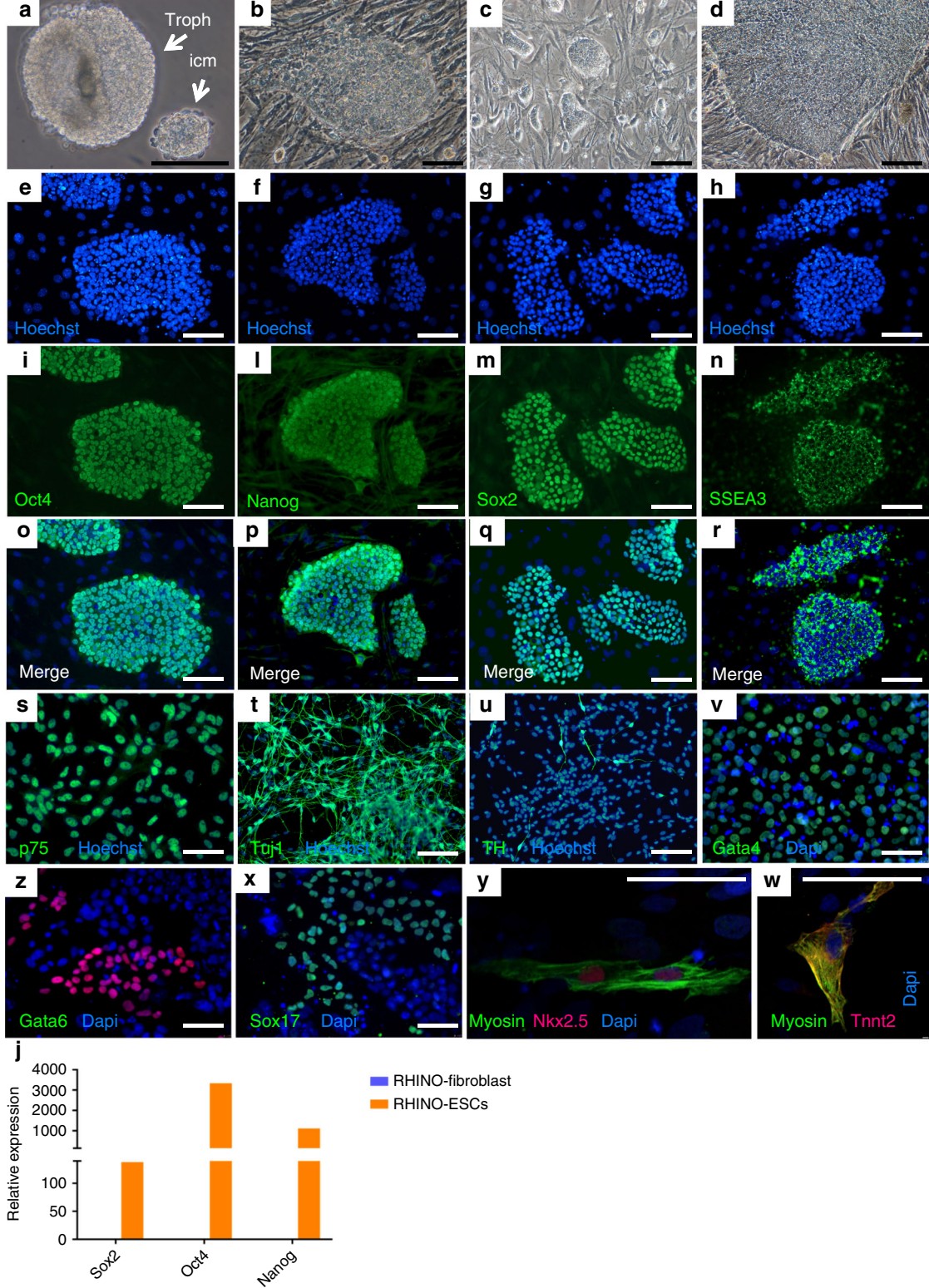

**Fig. 3** Isolation of inner cell mass and derivation/characterization of embryonic stem cell lines. **a** Isolated ICM and trophoblast from rhinoceros embryo after mechanical dissection with insulin needles (40×); **b** ICM outgrowth 48 h after plating on feeder cells (20×). **c** Representative image of ES colonies 24 h after enzymatic passaging; **d** ES colonies ready for the next passaging. Characterization of undifferentiated ES colonies: **e**, **i**, **o** *OCT4* staining; **f**, **l**, **p** *NANOG* staining; **g**, **m**, **q** *SOX2* staining; **h**, **n**, **r** *SSEA3* staining. Characterization of differentiated derivatives: **s** p75 staining of neural crest precursors; **t**, **u** Tuj1 and *TH* staining of neuronal cells; **v**, **z**, **x** *GATA4, GATA6, SOX17* staining of endodermal cells; **y**, **w** *MYOSIN, NKH2.5, TNNT2* staining of myocardial cells. Bar 100 μm; **j** RT-PCR for *SOX2, OCT4,* and *NANOG* in fibroblasts and rESC (*n* = 1 sample for each cell line)

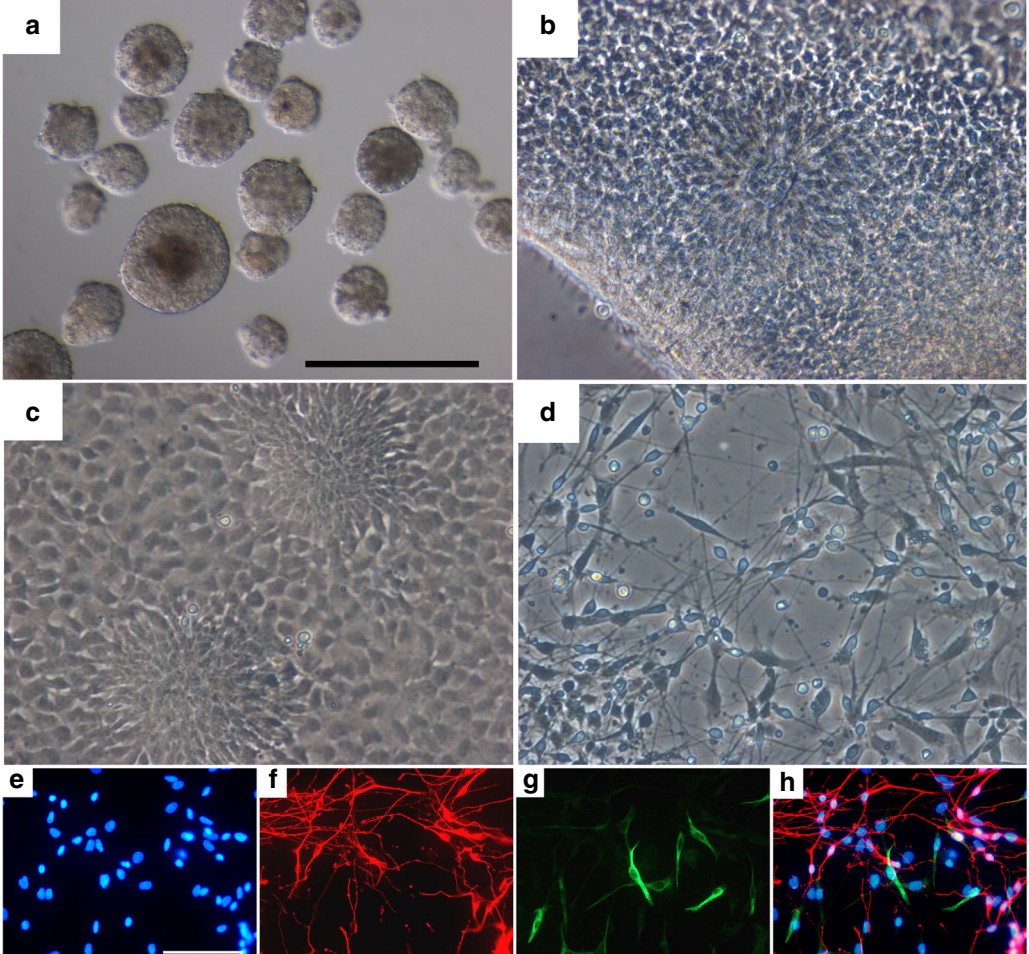

**Fig. 4** Bright-field panel of neural differentiation. **a** Embryoid bodies, 4 days after derivation from proliferating rESC. **b** Neural rosettes. **c** Proliferating neural precursors with areas of residual radial cells. **d** Differentiated neurons. Scale bar corresponds to 100 μm. Staining for neural differentiation obtained by embryoid body formation and subsequent induction of differentiation. **e** Hoechst staining of the nuclei, **f** differentiated neurons stained with *TUJ1*, **g** Peripherin staining, and **h** merge (see also ref. [21])

cells. However, we were not successful in doing so with SWR × NWR embryos due to their lack of a well-formed ICM, probably because they were plated 2 days earlier, and the fact that their outgrowths were mainly of trophoblast origin. These exhibited limited proliferation, sufficient only for a genetic analysis to confirm sex (one male and one female) and parentage (Supplementary Table 5).

The two successfully established cell lines (051A and 051B) had normal karyotype (Supplementary Figure 7) displaying a typical ES morphology of tightly packed cells, with a high nucleus/cytoplasm ratio, exhibiting growth in colonies on top of feeder cells with well-defined borders. The ES cell phenotype was confirmed by immunocytochemistry based on the pluripotency markers *OCT4*, *NANOG*, *SOX2*, and *SSEA3* and by real-time PCR (Fig. 3j). *OCT4* and *SOX2*-specific staining was also confirmed with monoclonal antibody staining (Supplementary Figure 4). Negative controls for unspecific binding of primary or secondary antibodies have been ruled out (Supplementary Figure 5). To further confirm the nature of true embryonic stem cells, the two established ES cell lines were induced to differentiate into neural, mesoderm, and endoderm derivatives. Differentiation towards the three lineages has been documented with at least three markers per lineage (Fig. 3). Neural differentiation through the formation of embryoid bodies and neural rosettes resulted in

*TuJ1* and Peripherin-positive cells (Fig. 4). From mesodermal differentiation beating cardiomyocytes were obtained (Supplementary Movie 1). Negative controls for unspecific binding of secondary to undifferentiated cells or primary antibodies to fibroblasts have been ruled out (Supplementary Figures 5 and 6 respectively).

### Discussion

Here we report the first ever documented generation of blastocyst, pre-implantation rhinoceros embryos through the application of assisted reproduction procedures that had never been previously attempted in this species. Our results suggest that these methods could play a valuable role in the effort to save rhinoceros populations on the brink of extinction. We also obtained embryos from SWR oocytes fertilized with semen of NWR males, clearly indicating the potential of rescuing genes of the NWR sub-species through intra-specific hybrid breeding. Moreover, we report the generation of ES cells with all the features of undifferentiated cells and a high capacity for differentiation in the three lineages. The next challenges will be to transfer the cryopreserved embryos into synchronized fSWR surrogate mothers to establish and carry a pregnancy to term. In parallel, we plan to harvest oocytes from the last two remaining fNWR in Kenya. Currently efforts are

underway to achieve the differentiation of rES cells to obtain artificially generated oocytes from banked somatic cells of several NWR individuals using iPS technology[4], as a means of achieving sustainable wide genetic diversity.

## Methods

**Ovarian stimulation protocol**. fSWR at an unknown stage of their estrus cycle were treated every other day with 3 ml of GnRH slow release (Histrelin® BioRelease, 0.5 mg/ml, Bet Pharm LLC, USA) for three or four times. Oocyte retrieval by OPU was performed 24 or 48 h after the last slow-release GnRH injection as reported in Supplementary Figure 1. This study was approved by the Internal Committee for Ethics and Animal Welfare of the IZW (approval no. 2015-03-03).

**Anesthetic procedures for oocyte recovery**. Serial ultrasonographical reproductive assessments and OPU procedures required safe immobilization and anesthesia protocols in rhinoceroses. A combination of butorphanol tartrate (50 mg/animal), detomidine hydrochloride (30 mg/animal), midazolam hydrochloride (20 mg/animal), and ketamine hydrochloride (200 mg/animal) was administered intramuscularly via dart for induction. The anesthesia was maintained by intravenous administration of 500 mg ketamine hydrochloride, 50 mg butorphanol tartrate, 10 mg detomidine hydrochloride, and 5 mg midazolam hydrochloride diluted in 500 ml 0.9% sodium chloride over a period of 30–60 min (CIR = 15–35 ml/min). Subsequently, anesthesia was reversed using atipamezole hydrochloride (70–100 mg/animal) and naltrexone hydrochloride (60–80 mg/animal)[23].

**Oocyte recovery by transrectal OPU**. Due to the large anatomical distance of more than 500 mm between cranial vaginal wall and the ovaries located closely to the kidneys in white rhinoceroses[1], the application of the standard transvaginal oocyte retrieval was not possible. Therefore, a new approach was developed performing the oocyte recovery through the cleaned and disinfected rectum. A 1500 mm long instrument (patent pending) guided the double lumen needle (inner diameter 1.4 mm outer diameter 2.1 mm) in an angle to the instrument axis of 85°. The needle was advanced directly through the rectal wall into the adjacent ovary located cranially to the anus at a distance of 1200–1400 mm. The angled needle could be advanced straight over a distance of up to 80 mm into the ovarian tissue. The aspiration process was completely monitored to the depth of 150 mm by an integrated 3D/4D real-time ultrasound transducer (5.0–9.0 MHz, RNA5-9-RS operated by portable ultrasound unit Voluson i, GE Healthcare, Germany). Each superovulated ovary contained up to 20 follicles on day 7 (Fig. 1). In general, follicles larger than 10 mm were selected, aspirated, and refilled to the original size for up to ten times by using a standard bovine flushing solution (Euroflush, IMV, France). The solution was flushed through the outer needle channel followed by repeated complete aspirations of the fluid through the inner needle channel using an aspiration unit (Cook Medical, Germany) adjusted to 100 mm Hg into a closed collection system. Flushing medium was heparinized with 20 IU heparin (Ratiopharm, Germany) per ml and together with the aspiration fluid were kept at 37 °C and collected into a sterile 50-ml falcon tube (Falcon, Germany) sealed with a silicon plug containing the needle connections and vacuum tube (Gynetics, Belgium). The aspiration fluid was searched for oocytes using a 70 μm nylon mesh filter-system (Falcon cell strainer) and a stereomicroscope (Carl Zeiss Microscopy, Germany). The oocytes were collected and washed in tissue culture plates (Falcon, Germany) followed by the transfer of each oocyte into a separate 2 ml sterile Nalgene® cryogenic vial (Merck, Germany). The vials filled with holding solution (H-SOF, Hepes-Synthetic Oviductal Fluid) were positioned into a portable battery-driven oocyte IVF embryo shipping incubator (Micro Q Technologies LLC, USA) which maintained a shipment temperature of 22 °C for up to 48 h.

**Sperm recovery and cryopreservation**. Semen was collected by electroejaculation (Seager model 14, Dalzell USA Medical Systems, USA) from eight captive rhinoceros bulls ($n = 5$ SWR, $n = 3$ NWR) using a hand-held stimulation probe[24] originally developed for elephants. Semen was collected into foam-insulated 50 ml isotherm collection tubes and kept at body temperature until analyzed. Concentration of spermatozoa in sperm-rich fractions was estimated immediately after collection using an improved Neubauer haemocytometer. The semen was then diluted to a concentration of ~$100 \times 10^6$ spermatozoa/ml using different, pre-warmed (36.5 °C) semen extenders: (1) Berliner Cryomedia containing 6% DMSO (v/v)[11,25,26] for extending NWR sperm; (2) BotuCrio, (Nidacon, Sweden) for extending SWR sperm. Total and progressive sperm motility was assessed after dilution and immediately post thawing. For this, 10 μl aliquots were put on a pre-warmed slide and evaluated on a warm stage equipped phase contrast microscope (Olympus C41; Olympus, Hamburg, Germany). From the fresh diluted and the frozen thawed semen samples, 10 μL aliquots were fixed in 40 μL Hancock's fixative for the assessment of acrosome integrity and sperm morphology[25,26]. Acrosomes were classified as intact versus modified or reacted (including completely detached acrosomes). Sperm morphology included search for a wide range of abnormalities. In addition, 10 μl aliquots were stained for 30 s in 40 μl of a one-step eosin–nigrosin stain (Sperm Vitalstain, Nidacon, Sweden) for assessment of sperm viability[27]. For viability assessment, the percentage of live

(unstained) and dead (stained) sperm cells was evaluated. For acrosome integrity, morphology, and viability, a total of 100 spermatozoa were evaluated per slide. Throughout the study all samples were evaluated by the same experienced spermatologist. Diluted white rhinoceros semen equilibrated for 15 min at room temperature before being packaged into 0.5 ml straws and chilled for 45/120 min to 4 °C (0.5 °C/min). Straws were frozen for 10 min at 3 cm above liquid nitrogen before being plunged into the liquid nitrogen. Freezing rates from +4 °C to −15 °C at 3 cm above LN surface were −11 °C/min. Freezing rates from −15 to −100 °C at 3 cm were −29 °C/min. Manual seeding was performed 60 s after freezing had started.

**In vitro maturation an ICSI of oocytes and embryo culture to blastocyst**. After recovery, oocytes were held at 22 °C in H-SOF media for 24–36 h during shipping. When delivered to the laboratory they were transferred to maturation media composed of DMEM-F12 with 10% rhinoceros estrum serum at 37.5 °C in a humidified atmosphere containing 5% of $CO_2$ in air[8] for 36–48 h. The timing required for in vitro maturation was determined empirically in preliminary experiments. After 36 h of maturation, the oocytes were treated with hyaluronidase and mechanically denuded with a pipette. Those that had a polar body (PB) were injected with sperm (ICSI), the others were returned to maturation and checked at regular intervals for the extrusion of the PB to be subsequently injected. Frozen semen of the selected bulls was used and live spermatozoa were separated by density gradient centrifugation (Redigrad 1 ml 90% and 1 ml 45%). Straws were thawed in water at room temperature and 100 μl of semen were placed on the top of the gradient and centrifuged at 700 g for 20 min. The pellet was transferred to a new centrifuge tube in 5 ml of TALP (Tyrode, Albumin, Lactate, Pyruvate) calcium-free media without albumin but supplemented with 1 mg/ml of polyvinyl alcohol and centrifuged again for 10 min at 300 g. The sperm pellet was re-suspended in mSOF IVF medium and diluted 1:1 in PVP 10% in H-SOF just before injection[28]. Oocytes were fertilized by ICSI using a piezo-driven micromanipulator (Prime Tech, Tokyo, Japan). Single motile spermatozoa with normal morphology were immobilized by piezo pulses and injected into the cytoplasm after piezo drilling to cut the zona pellucida and penetrate the oolemma. Injected oocytes (day 0) were cultured in modified SOF medium supplemented with BSA and MEM amino acids[28]. Half of the medium was changed on day 4, 6, 8, 10, and 12. Cleavage rate was evaluated 48 h post injection and blastocyst development at days 11–15.

Oocytes injected with NWR sperm were artificially activated within 1 h after ICSI. The oocytes were washed in a 0.3 M mannitol solution containing calcium (0.1 mM) and magnesium (0.05 mM), then transferred to the chamber of a cell fusion machine (SpyZot, LPS electronics, Cremona, Italy), subjected to two DC pulses of 1.5 kV/cm for 30 μm seconds in the same mannitol solution and then returned to mSOF media for in vitro culture at 37.5 °C in 5% $CO_2$ and 5% $O_2$[15].

**Embryo cryopreservation**. Expanding blastocysts on D11 to 13 with a diameter <200 μm with a clear detectable ICM were cryopreserved using a slow freezing protocol used for cattle and horse embryos. The embryos were equilibrated at room temperature for 5 min in a solution of 5% glycerol in H-SOF media, then 20 min in 10% glycerol. During equilibration, embryos were loaded into plastic straws (0.25 ml, three columns of media separated by air bubbles and the embryos loaded in the middle column). The freezing machine (Biocool IV, FTS Systems, USA; methanol bath) was pre-cooled at −6 °C. The straws were inserted into the methanol bath and after 5 min equilibration seeding was induced by touching the straw away from the central column with forceps pre-cooled in liquid nitrogen. After 10 min, when the entire straw was frozen with visible ice formed, the freezing program was activated with a cooling ramp of 0.5 °C/min until −32 °C. The straw was then plunged into liquid nitrogen and transferred to a liquid nitrogen tank for long-term storage.

**Stem cell derivation and characterization**. Embryos from fSWR × mSWR and fSWR × mNWR were cultured up to the expanded blastocyst stage to allow the formation of a well-defined ICM. In order to facilitate the process of expansion, the zona pellucida of the two SWR embryos and of one SWR × NWR embryo were removed by pronase treatment on day 12. The second SWR × NWR embryo had a very thin zona and hatched spontaneously on day 11. Following pronase treatment the embryos were allowed to re-expand overnight and then the ICM was mechanically separated from the trophoblast using insulin needles. The isolated cell masses were plated on mitomycin inactivated mouse fibroblasts seeded at a concentration of 50,000 cells per cm². The day of plating was D14 and D15 for the first and second SWR embryo respectively and D12 and D13 for the SWR × NWR embryos. The culture medium was DMEM-F12 supplemented with 20% KSR (Knockout Serum Replacement, Life Technologies), 2 mM glutamine (L-alanyl-L-glutamine, Sigma-Aldrich), 1% non-essential amino acids (Gibco), 0.055 mM beta-mercaptoethanol (Gibco), 10 ng/ml bFGF (Peprotech, UK) in 5% $CO_2$, 5% $O_2$ at 37.5 °C. Seven to nine days after plating the isolated ICMs were detached from the feeder cells by short exposure of five minutes to collagenase IV (Gibco) added to the culture medium at a final concentration of 1 mg/ml. The detached outgrowths were transferred in a glass petri dish and cut in 20–30 pieces using insulin needles and re-plated on fresh feeders. The mechanical passage was repeated a second time for all the embryos. The undifferentiated colonies derived from the

mechanical passaging of the two SWR embryos were subsequently exposed to a diluted trypsin solution (0.05%) for 5 min, then gently pipetted to obtain small cellular clumps, resuspended in complete culture medium, centrifuged and re-plated on fresh feeders. After enzymatic passaging, the newly formed colonies were readily visible within 24 h indicating high viability. The rES cells were subsequently expanded every 3–4 days with a dilution factor of 1:3. The established cell lines were characterized by a typical ES cell morphology of tightly packed cells, with high nucleus-cytoplasm ratio, growing as colonies with well-defined borders on top of feeder cells. Expression of ES markers OCT4 (#ab18976, Abcam, UK, dil 1:50, Mouse monoclonal, Santa Cruz, Oct3/4 sc-5279 dil 1:100), SOX2 (N1C3, #GTX101507, Genetex, USA, dil 1:200, Mouse monoclonal, Thermofisher, #MA-1-014 dil 1:100) and NANOG (N3C3, #GTX100863, Genetex, USA, dil 1:400), SSEA3 (MC-631, #MA1-020X, Thermofisher, USA, dil 1:200) were clearly demonstrated by immunohistochemistry. Under the same conditions, negative controls were performed on undifferentiated rES cells for the secondary antibody with the omission of the primary antibody (Supplementary Figure 5) and for primary antibodies on rhinoceros fibroblasts (Supplementary Figure 6) to rule out unspecific binding. RT-PCR was also performed for OCT4, NANOG, and SOX2. To quantify the gene expression, total RNA was extracted using RNeasy plus Mini Kit (Qiagen 74104). One micrograms of total RNA was converted into cDNA using the Superscript III cDNA synthesis kit (Thermofisher Scientific 18080093) as in manufacture's instructions. The quantitative real-time PCR was performed using the Quantitect-SYBR Green RT-PCR kit (Quiagen 204243) and the QuantStudio™ 6 Flex Real-Time PCR System (Thermofisher Scientific) for the relative quantification of genes of interest. The following were the thermal cycler conditions: activation 95 ℃ for 5 min, denaturation 95 ℃ for 30 s, annealing 60 ℃ for 30 s, and extension 72 ℃ (repeat 40 times). Triplicate Ct values were further analyzed ($2^{-\Delta\Delta CT}$) by normalizing to an endogenous reference gene (18srRNA). The list of primers can be found in Supplementary Table 2.

**In vitro differentiation of the three germ layers**. For neural differentiation[29], embryoid bodies (EBs) were generated from SWR rESC cultured on feeder cells at passages 5–7, when colonies were thick and clearly visible. The dishes were incubated with collagenase (1 mg/ml Collagenase Type IV, Gibco) for 20 min, then the detaching colonies were recovered and cultured in suspension in non-adherent dishes for 4 days in the same medium as the rESC but without bFGF. For generation of neural rosettes, EBs were plated in Matrigel Coated dishes (1:100 dilution, Matrigel BD) and cultured for 15 days in DMEM-F12 supplemented with glucose (0.6 mg/ml), 3 mM sodium bicarbonate, 5 mM HEPES, 2 mM glutamine, 100 µg/ml transferrin, 25 µg/ml insulin, 60 µM putrescine, 20 nM progesterone, 30 nM sodium selenite, 2 µg/ml heparin (all purchased from Sigma-Aldrich), and 20 ng/ml bFGF. Neural rosettes were mechanically isolated, cut into small pieces and cultured in the same media as above, but supplemented with 10 ng/ml of bFGF and 20 ng/ml of EGF to generate a population of proliferating neural precursors mostly p75 positive (#SC271708, Santa Cruz Biotechnologies, dil. 1:100) and cultured for at least 10 passages. To induce neural differentiation, growth factors were withdrawn from the culture media and cells were left to differentiate for at least 21 days in presence of ascorbic acid. After this time a network of neuron-specific BTubIII-(#A01627, GenScript, dil. 1:100) positive cells developed, a few of these neurons were also stained by TH (tyrosine hydroxylase) antibody (#P40101, Pelfreez, dil. 1:100) while other cells were positive to peripherin (#sc-377093 Santa Cruz, dil 1:100). (Figs. 3 and 4). Induction of the mesoderm differentiation was performed like described in Burridge et al.[30] between passage 15 and 25. In brief $4 \times 10^5$ SWR rES cells were seeded as single cell supplemented with 10 µM Rock inhibitor (#S1049, SelleckChem) on one well of a Geltrex-coated 6-well plate in MEF conditioned media (R&D System AR005) with 10 ng/ml FGF 2 for 3–4 days until the cells were 90–100% confluent. Thereafter the GSK 3 inhibitor CHIR99021 (#72054, 6 µM, Stemcell Technologies) was used to activate the WNT signaling pathway and thereby the cardiac differentiation of the cells. Two days later the media was replaced with RPMI with CDM3 supplemented with 5 µM IWP2 (# 72122, Stemcell Technologies). Beating cells were observed earliest 10 days later as small clusters.

Endodermal differentiation was performed either using the STEMdiff™ Definitive Endoderm Kit from Stemcell Technologies (# 05110) or using Knockout DMEM Knockout serum replacement media, (20% knockout replacement serum, 1% Glutamax, 1% nonessential amino acids, 0.1% β-mercaptoethanol, with 100 ng/ml Activin A (R&D Systems) and 0.1 ng/ml Wnt3 (Day1 only) (R&D Systems). The differentiation media was added (2 ml) and the media was changed every other day for 7 days.

SWR rESC-cardiomyocytes and primitive endoderm cells were seeded in a 24-well plate fixed, permeabilized, and blocked using the kit of Thermo Fisher Scientific (Human Cardiomyocyte Immunocytochemistry Kit, #A25973). For the cardiac staining, the antibodies within the Kit were used and complemented with the Myosin antibody (#10906-1-AP Proteintech, dil 1:100). The endodermal differentiation was validated using the following antibodies at 1:100 dilutions: GATA 4 (SantaCruz #sc-25310), GATA6 (#sc-9055, SantaCruz), Sox17 (#AF1924, R&D). Samples were analyzed using a LSM 510 Meta inverted confocal microscope (Carl Zeiss) and ZEN software (Carl Zeiss) and the LEICA DIMi8 and LAX Software.

**Parentage testing and sex determination**. For the two established rESC lines, cells (fSWR × mSWR) at passage 6–7 cells were pelleted after tryspinization in an eppendorf tube. For the two hybrid embryos (fSWR × mNWR), outgrowth of trophblast and differentiated ICM-derived cells were detached from the feeder cells by short exposure of 5 min to collagenase IV (Gibco) added to the culture medium at a final concentration of 1 mg/ml. The detached outgrowths were pooled and transferred to an eppendorf tube with a minimal volume of media. The eppendorf tube was centrifuged and the supernatant was discarded, stored at −80 ℃, and shipped in dry ice to the genotyping laboratory. DNA was extracted using peqGold Tissue DNA Mini Kit (Peqlab, Germany #12-3396-02) following the manufacturer instructions. From the hybrid embryos, we quantified the total amount of DNA in being 170 ng for embryos 240A equivalent to 28,000 cells and 285 ng from embryo 240B equivalent to 47,000 cells. PCR was conducted with the Type-it Microsatellite PCR Kit (Qiagen) according to manufacturer instructions but running every marker in a single 12.5 µl reaction. The touchdown protocol (63–55 ℃ or 58–50 ℃ with 2 ℃ lowering for every of the first four cycles) was executed for 25–35 cycles on a peqSTAR universal 96 gradient cycler (Peqlab, Germany). Fragment length analysis was processed on a 3130xl Genetic Analyzer (Applied Biosystems) with built in Genotyper software for allele discrimination. Markers, dye labels, and references are listed in Supplementary Tables 3 and 4. The analysis of the four samples was performed two times.

**Karyotype analysis**. The karyotype of the rhinoceros ESCs (between p6 and p16) was performed by the Laboratory for Human Genetics in Berlin. In brief, metaphase chromosomes were obtained from rESCs following the standard protocol for monolayer cultures and by treating the cells for 2 h with colcemid. Thereafter, the G banding of the mitotic cell was performed. In total, two passages of the rhinoceros ESCs were screened and 20 metaphases per passage were analyzed. Afterwards 18 karyograms were generated (standard diagnostic procedure for humans is to analyze 20 metaphases and to produce 5 karyograms) and no abnormalities in terms of clonal, structural, or numeric chromosome aberrations were detected. Chromosomes were identified and numbered according to previous reports found in Houck et al.[31]. Karyotypes were documented with the Axio Imager Z2 from Zeiss 630× and assembled using the Metafer and Ikaros software from MetaSystems (Supplementary Figure 7).

**Data availability**. All data supporting the results of this work are available within the paper and Supplementary Information.

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

## Acknowledgements

The authors acknowledge the ZOO Dvůr Králové, Avantea and the Nadace ČEZ for logistic and financial support. We thank our colleagues Arn Ludwig, Dietmar Lieckfeldt, Alessandra Quaggio (IZW), Irina Lagutina, Paola Turini, Andrea Perota (Avantea), Ejona Rusha (Helmholtz Zentrum München), Anje Sporbert, Narasimha Swamy Telugu, Norman Krüger (Max Delbrueck Center), Elke Schuemann, Gundula Thiel, Arno Schnorrenberg, Ivo Weissmann, and Peter Steiner for technical assistance. Steven Seet (IZW) for his activities in regards to fund raising, networking and public relation work for this project. We are thankful to numerous European zoological institutions (listed in alphabetic institutional order) for participating in the advanced assisted breeding program in rhinoceroses: Zoo Budapest, Zoo Chorzów, Zoo Dvůr Kralové, Longleat Safari Park, Montpellier Zoo, Zoo Pairi Daiza, Zoo Poznán, Zoo Salzburg, Zoo Schwerin, Zoo Thoiry. K.H. is supported by KAKEN 15K21736, 25114006, and 17H01395. P.L. is supported by H2020 EU Drynet and EraofArt. We acknowledge the slaughterhouses (L Scordamaglia, CEO of Inalca Spa, Franco Caffi, Vice-President of Prosus Spa and Maria Grazia Ragazzi of Zerbini & Ragazzi SNC) for access to key biological materials. We thank Ms. Brunella Gadda of DHL Express and her team with the logistic of shipping the rhino oocytes. We also thank our colleagues from San Diego Zoo Global for constructive discussions.

## Author contributions

T.B.H. and R.H. conceived the project, designed and performed the experiments, interpreted the data, coordinated the work. S.C. and S.D. share second authorship; S.C., S.D., F.G., G.L., J.S., and S.H. performed the experiments, collected and analyzed the data. M.B.R., K.H., P.L., and M.D. provided advice for experimental design. T.B.H., M.B.R., and S.D. revised the paper, C.G. conceived the project, designed the experiments, supervised the work, analysed and interpreted the data, wrote the paper. All authors jointly contributed to the final version of the manuscript.

## Additional information

**Competing interests:** The authors declare no competing interests.

