## [Peer Review File · Nature Communications]

Reviewers' comments:

Reviewer #1 (Remarks to the Author):

In this paper, the authors report the production of rhinoceros blastocysts following ovum pick up and ICSI. This is the first time rhinoceros embryo development to the blastocyst stage has been reported. The authors go on to produce what appear to be ES cells from the ICMs of these embryos, another significant advancement of ART in rhinos. Additionally, the authors report the successful production of intra-specific hybrid embryos and present their formation as the solution to saving the Northern white rhino sub-species from extinction. The advancements in rhinoceros ART presented in this paper are impressive, but the authors make many exaggerated claims regarding these accomplishments, in some cases, to the point of making statements that simply are not true. The viability of the intra-specific hybrid embryos is also suspect. More specific comments follow.

Primary comments:

1) Are claims supported by study results?

a. The authors' claim "Our results indicate that ART is not only a viable strategy to rescue the iconic, almost extinct, rhinoceros, but suggests that these techniques are applicable to other endangered mammalian species." is hyperbole and inappropriate in a respectable scientific journal. The authors have made some commendable progress, but they need to be more conservative in their interpretation of their results and they need to stick to the scientific facts and not exaggerate what they accomplished.

b. A point that gets lost is that the Northern white rhino is a sub-species of the Southern white rhino. Therefore, all references to "saving a species" are inaccurate. The loss of the Northern white rhino definitely is a conservation tragedy. However, the tragedy of losing a sub-species is not equivalent to losing an entire species.

c. The viability of the intra-specific hybrid embryos is suspect. The blastocysts did not form well-defined ICMs, and cell lines could not be established from them. The oocytes were artificially activated which often leads to parthenogenetic cleavage, and the authors do not report any control data for parthenogenesis. Although the parentage data presented in the table is rather convincing, detail on methodology is lacking, and the potential for embryo mosaicism is not addressed and seems plausible.

d. By producing intra-specific hybrid embryos, the authors have demonstrated the potential for saving genes of a sub-species that is almost extinct. This is a significant achievement, but stating it will save the sub-species from extinction is a leap of faith. No offspring have been produced with these embryos, and considering even the natural breeding program for the Northern white rhino sub-species inexplicably failed, it is unlikely that a program based on ART will fare much better. To my knowledge, no wildlife species has been saved by ART alone in the absence of natural breeding. If a few intra-specific hybrid calves were eventually produced, inbreeding would then be required to dilute the Southern white rhino genes enough to claim the Northern white sub-species had returned. Such severe inbreeding with few founders would likely lead to homozygous deleterious alleles. One example of this sub-specific hybridization strategy for saving a sub-species occurred in the

U.S. in Florida where Western pumas were introduced to the highly inbred Florida puma population to generate some hybrid vigor. After a few years, the Western pumas were removed so that the Western puma genes would start to get diluted with subsequent generations of breeding. The population is being monitored to determine if the strategy will work long-term or if the developmental defects of inbreeding will return.

2) Is the research unique and does it demonstrate significant advancement in the field?

The techniques used herein are not particularly novel or advancements of already existing and previously published techniques. The ovum pick-up procedure had to be customized for the anatomy of the rhino, but it has been published previously (Hermes, R., et al., 2009. *Theriogenology*, 72(7):959-968). The IVM/ICSI procedures were derived from those used in horses. Inter-specific IVF has been used for decades to evaluate the fertility of endangered species' sperm so the success of pig x rhino fertilization after ICSI or the success of intra-specific fertilization is no surprise. The intriguing fact and what makes this study commendable is that these technologies were applied to a rhinoceros with some success in producing blastocysts and ES cells. This is the first report of in vitro rhinoceros blastocyst and ES cell formation. Such information is valuable to the community of rhino researchers and conservationists. I leave it to the editors to decide if the use of existing technologies on a new species for the production of blastocysts and ES cells is worthy of publication in *Nature Communications*.

Specific comments:

Title: "stem cells from the endangered rhinoceros" stem cells were only generated for the Southern white rhino, a species that is not endangered.

Lines 32-35 awkward statement, because the horse and rhinoceros share a common ancestor, equine ART procedures may prove applicable to rhinos but I don't see how that relationship suggests Southern white rhinos should be involved.

Line 42 What is meant by "confirming the viability of these rhinoceros embryos"?

Line 44-46: Simply hyperbole – delete.

Line 49: Change "oocyte collection" to "ovum pick up".

Line 51-53: Awkward statement and it is a rhinoceros sub-species, not species.

Line 56: Be more conservative in your statements. Change "would ensure" to "could help to ensure".

Line 62: Typo - change NRWs to NWRs

Line 63: Typo - delete "this species" (redundant clause)

Lines 64-68: There are many different types of ART. The stated steps don't apply to all of them, just the strategy the authors chose to pursue.

Line 70: Usually when ovarian stimulation protocols are developed, oocytes are mature when aspirated (or very close to it). That was not the case in this study so the protocol seemed ineffective.

Line 87-90: Did the investigators have control oocytes that were pulsed to determine how many would cleave parthenogenetically and develop to blastocysts in response to this treatment? I realize they would not use rhino oocytes but maybe horse or pig oocytes?

There are numerous publications dating way back regarding the development of oocytes to

blastocysts following activation, and sometimes a fairly high percentage of oocytes do so see as examples:

BIOLOGY OF REPRODUCTION 58, 1177-1187 (1998) Development of Parthenogenetic and Cloned Ovine Embryos: Effect of Activation Protocols' P. Loi,^{2,3} S. Ledda,⁴ J. Fulka, Jr., P. Cappai, ³ and R.M. Moor⁶;
Development 109, 117-127(1990) Printed in Great Britain © The Company of Biologists Limited 1990 The parthenogenetic development of rabbit oocytes after repetitive pulsatile electrical stimulation JEAN PIERRE OZIL).

Lines 102-105: Oocytes were held for 60-80 hr prior to ICSI. This time frame seems more appropriate for immature oocytes retrieved post-mortem without any ovarian stimulation protocol prior to collection. Perhaps the stimulation protocols had no effect? how do you explain the lack of maturity?

Lines 132-134: It is plausible that the developmental difference between SWR x SWR embryos and NWR x SWR embryos is an indication that the intra-specific hybrid embryos are unhealthy or that some of them are parthenotes.

Lines 147-150: Hyperbole - suggest changing to "Our results indicate that these methods may be useful for rescuing genes of the NWR sub-species through intra-specific hybrid breeding."

Supplementary information

Line 4-8; section 1) Question about the effectiveness of ovarian stimulation protocol already mentioned above.

Lines 72-97; section 5) Question regarding the length of IVM required and its indication that the oocytes are all immature at collection already mentioned above.

Lines 93-97; section 5) Were control pig oocytes subjected to pulses to determine parthenogenetic cleavage rates and development to blastocysts following such treatment?

Section 7) I am not an expert in validating ES cells but it would be prudent to include antibody dilutions used in the immunohistochemistry as well as negative controls. Were cells stained with secondary antibody in the absence of primary to prove binding specificity of fluorescence?

Section 8) It is not clear how long the cell lines had been maintained or how many passages they had undergone when the differentiation part of this project was conducted.

Section 9) It is not clear in this description what cells were used for parentage determination. Based on the text in the paper, it appears that trophoblast cells were used from the NWR x SWR intra-specific hybrid embryos. How do the authors rule out chimerism/mosaicism of these embryos? Because the ICM did not develop well like it did with SWR x SWR embryos, and because the oocytes were artificially activated by electrical pulses, is it possible that the ICM was made up of parthenogenetically dividing cells? More detail is required in this section. How many cells were analyzed? How many times was the analysis run? How did the results compare between SWR x SWR and NWR X SWR testing?

Section 10) Was there reference to karyotypes in this paper? I don't recall it being mentioned. Please include more detail regarding the number of cells examined, the proportion successfully karyotyped and any anomalies noted.

Figure S3 – In the heading, please describe the point of this figure. To demonstrate cells were diploid?

Table S3 – Parthenote control data really needs to be included especially following artificial activation.

Video – I could not play the video of the beating cells but I believe the authors when they say they observed them. I have seen the same occur when I cultured late stage equine blastocysts for longer periods.

Reviewer #2 (Remarks to the Author):

SUMMARY

This manuscript describes efforts to develop assisted reproductive technologies (ART) to rescue the northern white rhinoceros (NWR), an almost extinct species. Oocytes were harvested from southern white rhinos (SWR) by transrectal ovum pickup (OPU), matured, and fertilized by intracytoplasmic sperm injection (ICSI). The resultant embryos developed to the blastocyst stage. Next, hybrid rhino embryos were generated in vitro using NWR and SWR gametes. Cell lines were also established from blastocyst generated with SWR oocytes and NWR sperm that exhibited typical embryonic stem (ES) cell features. Some purebred and hybrid blastocysts were frozen for later transfer.

OVERALL COMMENTS

This manuscript details a tremendous effort to develop ART approaches with the goal of saving an endangered species. In general, the manuscript is well written with a logical set of experiments and quality methods and results. The major claims, experimental outcomes and their findings are justified by the results, novel and appropriately discussed. However, there is some question about whether the findings are of wide interest.

Major Comments:

(1) Do the authors have any experimental proof that the generated ES cell lines will contribute to an embryo when injected into a blastocyst? This experiment has to be done to establish that the derived ES cells will contribute to the germline and useful to regenerate individuals.

Reviewer #3 (Remarks to the Author):

This manuscript documents a huge effort in advancing assisted reproduction technology in rhinoceros. It clearly represents an exercise in teamwork by a number of knowledgeable experts. This kind of research represents the only hope of keeping some species of mammals from extinction. While success rates of some of the steps were on the low side,

photomicrographs of the inner cell masses of several blastocysts document excellent morphology. The health of these cells also is documented by the cell lines produced, lines that have many of the properties of bona fide embryonic stem cells. Whether these cell lines will lead to germ line transmission remains to be seen, but this appears to be likely, one way or another. That these cell lines are normal diploid in chromosome composition, with the sperm and oocyte contributing is well documented. Presumably the authors are thinking about how to get viable pregnancies from this material.

I have embarrassingly little to add to improve the manuscript. I do suggest the following rewording:

L 51- embryos

L 61- ...challenges; there are only...

L 62- ...bigger than the horse and

L 63- deleted duplicated this species

L 68- sources of stem cells

L 107- cleavage,

Reviewer #4 (Remarks to the Author):

This is certainly an interesting and timely report describing approaches to try to rescue an endangered species for which few individuals remain, namely the Northern White Rhino. Fortunately, in this case the closely related Southern White Rhino is currently much more numerous and available to help in the project. In the paper the authors report on several approaches including in vitro fertilization using intracytoplasmic sperm injection (ICSI), and also the derivation of embryonic stem (ES) cells, which could potentially be used in the future to produce artificial gametes. Such ES cells could also provide a gold standard against which to characterize iPS cells if they are derived in the future. Clearly the authors are working in a 'non-ideal' situation with only two female and one Northern White Rhino still surviving. However, they have successfully developed methods for producing rhino embryos to a blastocyst stage for the Southern White Rhino, and hybrid embryos from Southern White Rhino eggs fertilized with Northern White Rhino sperm. Unfortunately the quality of available Northern White Rhino eggs precluded obtaining embryos from them. In addition they were able to derive putative ES cell lines from the Southern White Rhino embryos.

These results seem to me to be a laudable first step. However, the authors might better discuss how they will be used going forward. Clearly hybrid embryos, if allowed to develop to adult rhinos, will not replicate the Northern White Rhino: what strategies do the authors envisage to recreate that species. Likewise how do they envisage using ES cells or iPS cells: at the moment iPS cells would be the only way to capture the whole Northern White Rhino genome, but they have yet to be made, and the ability to generate functional gametes also lies in the future. Nevertheless, the newly created rhino ES cells will provide an important reference point for future work with iPS cells. However, given the importance of these new lines, in my view, their current characterization is insufficient:

In Figure 3, pictures of these new ES cells are provided. However, as seems common in the

literature these days, the magnification is far too low to be able to make out the cells. ES cells have a fairly typical morphology – but that require high magnification pictures to see. In the same figure, immunostaining for Oct4, Nanog, Sox2 and the surface antigen SSEA3 is shown. However, first, no negative control staining is shown. Second, there is nothing to confirm the specificity of the antibodies: these are rabbit polyclonal antibodies, so the absence of staining by irrelevant heterotypic antibodies must be shown – at least in the case of the Abcam anti-Oct4 antibody, the artificial peptide used as an immunogen is available as a blocking peptide to test the specificity of staining. How do we know that these antibodies made to mouse proteins will cross react with the rhino equivalents? Further, as we know little about expression of these markers in rhino embryos, there is a leap of faith in assuming that their expression patterns in rhinos will match that in other species – a reasonable expectation but it should be demonstrated. In fact in the case of SSEA3, it is a good marker for human ES cells but it is not expressed by mouse ES cells: we clearly do not know how it will behave in rhinos. I would suggest that a better approach would be to collect data on expression of a wider range of genes, perhaps by RNAseq, which would then provide a better way to compare with ES cells of other species.

More importantly than characterizing marker expression is a functional demonstration of pluripotency. The generally accepted way to characterize a pluripotent stem cell is to show differentiation into derivatives of all three germ layers. In the mouse this is generally done by making chimeric embryos. This would clearly be a challenge in the rhino but the approach of making xenograft teratomas in immunodeficient mice is generally used for human ES cells and could easily be tried with the rhino ES cells. As it is, the authors have just done limited in vitro differentiation, assessed by immunostaining for just one marker for each germ layer (with the same caveats as above regarding lack of negative controls and evidence of antibody specificity).

Overall, in my view, the derivation of the rhino ES cells is an important result, but these lines deserve a more detailed characterization as summarized above.

Overall Reviewers' comments:

We thank all the reviewers for their constructive comments and helpful suggestions that have improved the manuscript.

Reviewer #1 (Remarks to the Author):

In this paper, the authors report the production of rhinoceros blastocysts following ovum pick up and ICSI. This is the first time rhinoceros embryo development to the blastocyst stage has been reported. The authors go on to produce what appear to be ES cells from the ICMs of these embryos, another significant advancement of ART in rhinos. Additionally, the authors report the successful production of intra-specific hybrid embryos and present their formation as the solution to saving the Northern white rhino sub-species from extinction. The advancements in rhinoceros ART presented in this paper are impressive, but the authors make many exaggerated claims regarding these accomplishments, in some cases, to the point of making statements that simply are not true. The viability of the intra-specific hybrid embryos is also suspect. More specific comments follow.

We are pleased that the reviewer finds our advancements in this paper impressive, but we do not feel we have made exaggerated claims. We now provide additional evidence below to show the viability of the hybrid embryos.

Primary comments:

1) Are claims supported by study results?

a. The authors' claim "Our results indicate that ART is not only a viable strategy to rescue the iconic, almost extinct, rhinoceros, but suggests that these techniques are applicable to other endangered mammalian species." is hyperbole and inappropriate in a respectable scientific journal. The authors have made some commendable progress, but they need to be more conservative in their interpretation of their results and they need to stick to the scientific facts and not exaggerate what they accomplished. R# We respectfully disagree with the referee. We (and we imagine others) are working with the same approach in the Sumatran Rhinoceros, in the Indian Rhinoceros, etc. and believe these techniques are applicable not only to these but also to other species such as the Bovidae (Gaur), the Cervidae and others, after the required species species-specific adaptations and refinements. However, we are happy to revise this sentence.

b. A point that gets lost is that the Northern white rhino is a sub-species of the Southern white rhino. Therefore, all references to "saving a species" are inaccurate. The loss of the Northern white rhino definitely is a conservation tragedy. However, the tragedy of losing a sub-species is not equivalent to losing an entire species.

R# We accept the point and have amended the text accordingly. However, some scientists do not agree with this, and the scientific opinion about the taxonomic status of the northern white rhino is not homogenous. The IUCN declares the NWR as a subspecies of the southern white rhinos. However, one of the leading taxonomists and primatologists, the British-Australian scientist Colin Groves and colleagues published a contrary opinion about the taxonomic status of the NWR¹. They declared the NWR as the 6th rhino species. Colin Groves, who is recognised for his groundbreaking research in the field of primatology, used morphological and genetic techniques to revise the classical rhinoceros taxonomy.

c. The viability of the intra-specific hybrid embryos is suspect. The blastocysts did not form well-defined ICMs, and cell lines could not be established from them. The oocytes were artificially activated which often leads to parthenogenetic cleavage, and the authors do not report any control data for parthenogenesis. Although the parentage data presented in the table is rather convincing, detail on methodology is lacking, and the potential for embryo mosaicism is not addressed and seems plausible. R# The hybrid embryo pictured in fig 2 i would challenge any embryologist to conclude that this is **not** a very good embryo. In addition, reviewer 3 states that it the blastocysts have “excellent morphology” We isolated 2 ES cell lines from 4 attempts. The fact that we did not achieve 4/4 is not “suspect” but is simply the nature of the experimentation. We kindly ask the reviewer to consider that we are working with a species in which it is not possible to harvest and work on an unlimited supply of oocytes. The electrical activation we performed is not sufficient to parthenogenetically activate a large mammal oocyte. What we have used here is an incomplete protocol that does not allow the retention of the second polar body (that is obtained by culturing for 3-6 hours in presence of cytochalasin B, a microtubule inhibitor and or 6-dimethyl amino purine) and if activation occurs it would result in an haploid embryo with a very low developmental competence². This protocol to assist embryo development after ICSI, defined as assisted oocyte activation (AOA), is used in human fertility clinics especially for male infertility factor (as in our case) and after embryo transfer normal babies have been generated^{3,4,5}. These references have been added to the paper. We also have performed additional experiments to demonstrate that parthenogenesis it is an unlikely event (data added to table S3 completed with the data not available for mNWR3 and pig oocytes subjected to the AOA with one electrical pulse). This was done with pig eggs since to make controls for parthenogenesis with rhinoceros eggs would be unethical. It is clear from the data that the pig oocytes do not retain the second polar body thus making them haploid and none of the activated one went beyond the cleavage stages. We were cognisant about the possibility of parthenogenesis and therefore we put in place a number of check points at different levels of development including DNA microsatellite analysis to exclude beyond any reasonable doubt that we are not generating parthenogenetic embryos. The potential for mosaicism was not assessed because for parentage testing you need a substantial amount of DNA and for the hybrid embryos we used **all** the embryo outgrowths and **all** of the cells constituting the embryos, including ICM derived ones that differentiated towards a trophoblast phenotype based on morphology. In addition, we sacrificed the half of the DNA that was store for future analysis of the 2 NWR embryos to repeat the microsatellite analysis as requested and these confirmed the earlier results. From the hybrid embryos we quantified the total amount of DNA in being 170 ng for embryos 240A equivalent to 28,000 cells and 285 ng from embryo 240B equivalent to 47,000 cells making embryo mosaicism unlikely.

d. By producing intra-specific hybrid embryos, the authors have demonstrated the potential for saving genes of a sub-species that is almost extinct. This is a significant achievement, but stating it will save the sub-species from extinction is a leap of faith. No offspring have been produced with these embryos, and considering even the natural breeding program for the Northern white rhino sub-species inexplicably failed, it is unlikely that a program based on ART will fare much better. To my knowledge, no wildlife species has been saved by ART alone in the absence of natural breeding. If a few intra-specific hybrid calves were eventually produced, inbreeding would then be required to dilute the Southern white rhino genes enough to claim the Northern white sub-species had returned. Such severe inbreeding with few founders would likely lead to homozygous deleterious alleles. One example of this sub-specific hybridization strategy for saving a sub-species occurred in the U.S. in Florida where Western pumas were introduced to the highly inbred Florida puma population to generate some hybrid vigor. After a few years, the Western pumas were removed so that the Western puma genes would start to get diluted with subsequent generations of breeding. The population is being

monitored to determine if the strategy will work long-term or if the developmental defects of inbreeding will return.

R# The reviewer is right, to date no species has been saved by the use of ART alone. Yet, in this particular case, ART is the only tool remaining to save this subspecies, as all other breeding options have been tried and failed. This is the rationale of our approach. We are well aware of inbreeding effects that this might bring, but our approach has been endorsed by the African Rhino Specialist Group (see letter attached). This new approach is the first of its kind and might be one used more often in the future to save a species from extinction when natural breeding has become an excluded option. We never claimed we would save this sub-species but we are proposing a science-based procedure with solid data. We are following the widely accepted strategy proposed in Ref 9, a white paper that carefully examined all possible options for conservation. There are somatic cells of at least a dozen NWR individuals available in cell banks. iPS cells have been already developed in rhinoceros (Ref 22 in revised paper) and mice have now been generated by artificial gametes derived from iPS cells (Ref 10). This is our long-term goal and what we are presenting here are some essential cornerstones of what is required. We agree it will be a long process and possibly not achievable, but our approach is providing novel data to underpin this aim.

2) Is the research unique and does it demonstrate significant advancement in the field?

The techniques used herein are not particularly novel or advancements of already existing and previously published techniques. The ovum pick-up procedure had to be customized for the anatomy of the rhino, but it has been published previously (Hermes, R., et al., 2009. *Theriogenology*, 72(7):959-968). The IVM/ICSI procedures were derived from those used in horses. Inter-specific IVF has been used for decades to evaluate the fertility of endangered species' sperm so the success of pig x rhino fertilization after ICSI or the success of intra-specific fertilization is no surprise. The intriguing fact and what makes this study commendable is that these technologies were applied to a rhinoceros with some success in producing blastocysts and ES cells. This is the first report of in vitro rhinoceros blastocyst and ES cell formation. Such information is valuable to the community of rhino researchers and conservationists. I leave it to the editors to decide if the use of existing technologies on a new species for the production of blastocysts and ES cells is worthy of publication in *Nature Communications*.

R# We thank the reviewer for acknowledging the achievements reported in this paper as unique. We agree with the reviewer that we are not using novel techniques, but most of the ground-breaking scientific literature does not use novel techniques. We therefore ask the editors to decide on the novelty of our work which presents a whole new approach towards species conservation when all other systems have failed to save a species from extinction. It brings new perspective not only to the small community of rhino researchers but to the biotechnology researchers as a whole who may not be aware of the importance of their (bio)technology techniques (e.g. production of artificial gametes) may have on species conservation in the future.

However please note that a different technique was used to harvest ova. Because of the size of the animals, a transrectal route has to be taken and for this we developed a 150 cm long modular OPU system which is operated from outside the rectum allowing a precise puncture, aspiration and flushing of follicles larger than 5 mm. The system is currently in the patenting process. In addition to this advance, we provide novel information on the ovarian physiology, on the timing of in vitro maturation, and on the embryo kinetics of the ability of SWR ICM cells to proliferate in vitro as putative ES cells. In table S4, we provide data for embryos obtained in 2017 after two unsuccessful years in 2015 and 2016 with no embryos.

Specific comments:

Title: “stem cells from the endangered rhinoceros” stem cells were only generated for the Southern white rhino, a species that is not endangered.

R# The SWR is currently rated as ‘near threatened’ by the IUCN Red List of Threatened Species™, and is at risk of becoming endangered. The NWR however is rated as ‘extinct in the wild’ a status far beyond being endangered. The title we chose is a bridge between those two official statuses of threat and cumulates best in our perspective results presented for SWR and SWR x NWR hybrid embryos. We can edit the title accordingly if the editor feels this is appropriate and there is no space limitation.

Lines 32-35 awkward statement, because the horse and rhinoceros share a common ancestor, equine ART procedures may prove applicable to rhinos but I don’t see how that relationship suggests Southern white rhinos should be involved.

R# SWR is the means to translate (test, develop, optimize, etc.) well established techniques to the NWR; moreover they will be required for the hybrid strategy.

Line 42 What is meant by “confirming the viability of these rhinoceros embryos”?

R# We agree that ES cells alone do not confirm the viability of rhinoceros embryos -the final definition of embryo viability is the ‘take home baby’ rate. We are starting to address the embryo transfer issue that it is not trivial considering the size of the animals (2,000 kg average), and when successful, will need 16 months pregnancy. So at this stage the best assessment of the viability of the embryos was to derive ESC lines. We know that for this you need good quality embryos and the 50% success rate (4 embryos 2 ES cell lines) we obtained speaks by itself. Moreover, we needed ESC lines to develop a protocol for making artificial gametes, as this will be the most promising route in the long run) and compare them with iPS cells that would eventually be the primary source to be used to make gametes to enhance genetic diversity. However, to clarify the statement we have inserted the word “potential” to clarify our meaning.

Line 44-46: Simply hyperbole – delete.

R# I report here the last sentence guidelines of Nature for the introductory paragraph “and finally, 2-3 sentences putting the main findings into general context so it is clear how the results described in the paper have moved the field forwards”. Therefore we have reworded the sentence to clarify.

Line 49: Change “oocyte collection” to “ovum pick up”.

R# changed

Line 51-53: Awkward statement and it is a rhinoceros sub-species, not species.

R# changed

Line 56: Be more conservative in your statements. Change “would ensure” to “could help to ensure”.

R# changed

Line 62: Typo - change NRWs to NWRs

R# changed

Line 63: Typo - delete “this species” (redundant clause)

R# changed

Lines 64-68: There are many different types of ART. The stated steps don't apply to all of them, just the strategy the authors chose to pursue.

R# We agree there are many type of ART, but here we describe the ones relevant to our wish to conserve NWRs. In these introductory statements, it is clear that we have not implanted embryos into foster mothers, as clearly indicated into the title.

Line 70: Usually when ovarian stimulation protocols are developed, oocytes are mature when aspirated (or very close to it). That was not the case in this study so the protocol seemed ineffective.

R# we respectfully disagree with the referee. Ovarian stimulation protocols in cattle, for example, are used not to recover mature oocytes but to increase the size and possible the quality of the follicles but without triggering the resumption of meiosis⁶. To trigger resumption of meiosis you need to administer HCG or GnRH when the follicle has reached a suitable size that varies according to the species. This is what it is currently done for human assisted reproduction but this is what we need to avoid in animals as we can not monitor them as it is done in IVF clinics for humans. Also collecting pre-ovulatory follicles is another technical challenge in large animals usually resulting in a low recovery rate. As it can be see in Fig 1d, all the cumulus oocyte complexes have a compact morphology indicating that they have not resumed meiosis and are all that the same developmental stage. This is what is normally done with livestock oocytes as they all will require all the same timing to reach metaphase II. This is essential because the oocytes had to be shipped in holding conditions awaiting initiation of maturation in vitro when they reach the laboratory.

Line 87-90: Did the investigators have control oocytes that were pulsed to determine how many would cleave parthenogenetically and develop to blastocysts in response to this treatment? I realize they would not use rhino oocytes but maybe horse or pig oocytes? There are numerous publications dating way back regarding the development of oocytes to blastocysts following activation, and sometimes a fairly high percentage of oocytes do so see as examples:

BIOLOGY OF REPRODUCTION 58, 1177-1187 (1998)Development of Parthenogenetic and Cloned Ovine Embryos: Effect of Activation Protocols' P. Loi,2,3 S. Ledda,4 J. Fulka, Jr., P. Cappai, 3 and R.M. Moor6;

Development 109, 117-127(1990) Printed in Great Britain © The Company of Biologists Limited 1990 The parthenogenetic development of rabbit oocytes after repetitive pulsatile electrical stimulation JEAN PIERRE OZIL).

R# see extensive response to point 1c above.

Lines 102-105: Oocytes were held for 60-80 hr prior to ICSI. This time frame seems more appropriate for immature oocytes retrieved post-mortem without any ovarian stimulation protocol prior to collection. Perhaps the stimulation protocols had no effect? how do you explain the lack of maturity?

R# As noted above the aim was to harvest immature oocytes and transport them by keeping them immature to the laboratory and then initiate in vitro maturation. The stimulation protocol did not intend to trigger oocyte maturation but only follicular development. Again we translated this holding procedure from the horse (see ref 3 and 16). By holding at 22°C the oocytes do not resume meiosis. This holding time for the rhinoceros oocytes lasted from 24 to 36 h. Oocytes maturation started when the oocytes reached the laboratory, were transferred in the maturation medium and incubated at 37.5 °C. Maturation time was optimized at 36 to 44. This timing is comparable to the equine or pig oocytes, only a few hours longer as it was the time required for blastocyst development. 38.6% mat II rate is not high, we agree, in the horse is 50%, in the pig 90%, but as a ground breaking work it is respectable, we

will hopefully improve it with time.

Lines 132-134: It is plausible that the developmental difference between SWR x SWR embryos and NWR x SWR embryos is an indication that the intra-specific hybrid embryos are unhealthy or that some of them are parthenotes.

R# In biology everything is possible but we have to stick to evidence based observations. There was a healthy birth of a hybrid (father SWR / mother NWR Nasima) at Dvur Kralove Zoo November, 11th, 1977. The female hybrid lived for 30 years.

Having said that and ruled out (100% with the DNA parentage test) that any of the hybrid embryos are parthenogenetic, it might be possible that these hybrids are less fertile as it is documented by the work done in the hybridization of the Swamp Buffalo (draft) with the Mediterranean buffalo (dairy) to replace one species with the other in thousands of farms in China and South East Asia to increase milk production⁷.

Lines 147-150: Hyperbole - suggest changing to “Our results indicate that these methods may be useful for rescuing genes of the NWR sub-species through intra-specific hybrid breeding.”

R# we respectfully disagree with the reviewer as there are no reasons why we could not harvest oocytes from the two female NWR left in Kenya, generate an embryo, a pregnancy and a baby. It will not be easy but possible. It is a race against time. If the publication of this work is not further delayed, it might convince the Kenyan Authorities to allow us to take the oocytes of the two remaining females back to our lab, since this is the first evidence that real, viable embryos of the Rhinoceros species can be made. The last male, Sudan, was euthanized on March 19th because of health problems, and eventually the next will be the turn of the 2 females.

Supplementary information

Supplementary information

Line 4-8; section 1) Question about the effectiveness of ovarian stimulation protocol already mentioned above.

R# already answered

Lines 72-97; section 5) Question regarding the length of IVM required and its indication that the oocytes are all immature at collection already mentioned above.

R# already answered

Lines 93-97; section 5) Were control pig oocytes subjected to pulses to determine parthenogenetic cleavage rates and development to blastocysts following such treatment?

R# answered extensively above at 1c.

Section 7) I am not an expert in validating ES cells but it would be prudent to include antibody dilutions used in the immunohistochemistry as well as negative controls. Were cells stained with secondary antibody in the absence of primary to prove binding specificity of fluorescence?

R# The ES cell staining was performed on ES cells growing on mouse feeder cells. While ES cells are strongly positive the feeder cells are clearly negative for OCT4, SOX2 and NANOG demonstrating the specificity of the antibody for undifferentiated nuclei. The same antibodies were used to stain the neural precursors derived from ES cells and in that case the staining was negative demonstrating that the secondary antibody per se cannot stain rhino cells. Moreover the nuclear staining of OCT4,

NANOG and SOX2 is very typical of ES cells because nuclei are intensively stained while nucleoli are negative. In all instances the underneath mouse feeders are negative. The antibody staining was performed in Cremona with the antibody presented in the SI but also duplicated in Berlin with a completely different source of antibody listed here (except for the SSEAs that was the same) with the same results. The antibodies were diluted 1:100. As negative control we used the same primary and secondary antibodies and stained rhinoceros fibroblasts without getting a signal when using the same exposure time used for the rESCs.

Oct-3/4 (H-134), rabbit polyclonal, Santa Cruz: sc-9081

- anti-OCT4, rabbit polyclonal, Life tech. Staining Kit; A24867
- Nanog, rabbit polyclonal, Thermo Fisher: PA1-097
- Nanog, goat polyclonal, R&D Systems: AF1997
- SOX2 (NH2 terminus), rabbit polyclonal, BioLegend: 630802
- anti-SOX2, rat monoclonal, Life tech. Staining Kit; Bestellnummer: A24759
- SSEA3 MC631; rat monoclonal, Thermo Fisher: MA1-020x
- SSEA4 MC-813-70; mouse monoclonal, Thermo Fisher: MA1-021x

The dilution of all antibodies were standard 1:100.

We have provided as an additional file (because we have already reached the limit of 10 display items indicated on the instruction to authors) the staining only with the secondary antibody in the absence of the primary to prove the specificity of the binding as requested as well as fibroblasts staining and a higher magnification of the undifferentiated rESC..

Section 8) It is not clear how long the cell lines had been maintained or how many passages they had undergone when the differentiation part of this project was conducted.

R# For mesodermal and endodermal differentiation all experiments were conducted between p15-p25.

In between the cells were maintained on inactivated mouse embryonic fibroblast (Global Stem, GSC-6001) supplemented with MEF conditioned media (R&D, AR005) with extra 10 ng/ml FGF2.

Regarding neural differentiation, EBs from which neural rosettes were derived were generated from rESC from passage 5 to 7. (Proliferating precursor's cells were cultured for at least 10 passages.

Terminal differentiation into a mixed population of neurons was performed at passage 3 and passage 7).

This information has been added to the SI file.

Section9) It is not clear in this description what cells were used for parentage determination. Based on the text in the paper, it appears that trophoblast cells were used from the NWR x SWR intra-specific hybrid embryos. How do the authors rule out chimerism/mosaicism of these embryos? Because the ICM did not develop well like it did with SWR x SWR embryos, and because the oocytes were artificially activated by electrical pulses, is it possible that the ICM was made up of parthenogenetically dividing cells? More detail is required in this section. How many cells were analyzed? How many times was the analysis run? How did the results compare between SWR x SWR and NWR X SWR testing?

R# We have answered this extensively in point 1c and repeated the microsatellite analysis as requested by the referee.

Section 10) Was there reference to karyotypes in this paper? I don't recall it being mentioned. Please include more detail regarding the number of cells examined, the proportion successfully karyotyped and any anomalies noted.

R# See ref 31 and also we consulted Atlas of mammalian chromosomes. (Eds) O'Brien SJ, Menninger JC, Nash WG (2006), John Wiley & Sons, Inc. Hoboken, NJ, USA, ISBN-13 978-0-471-35015-6,

ISBN-10 0-471-35015-X. Work was outsourced to a human reference laboratory as indicated. In total 2 passages of the rhinoceros ESCs were screened and 20 metaphase's per passage analyzed. Afterwards 18 karyograms were generated (standard diagnostic procedure for humans is to analyze 20 metaphases and to produce 5 karyograms) and no abnormalities in terms of clonal, structural or numeric chromosome aberrations were detected.

Figure S3 – In the heading, please describe the point of this figure. To demonstrate cells were diploid?
R# Done

Table S3 – Parthenote control data really needs to be included especially following artificial activation.
R# Done

Video – I could not play the video of the beating cells but I believe the authors when they say they observed them. I have seen the same occur when I cultured late stage equine blastocysts for longer periods.

R# We regret this but it is in a universal format that can be played by simple tools (<https://www.videolan.org/vlc/index.it.html>). It is true that beating cells can be obtained when cells spontaneously randomly differentiate. However it has to be noted that we got the cells after applying a cardiac specific differentiation protocol. Moreover we only wanted to demonstrate that the cells can differentiate into the mesodermal lineage using specific protocols. Other cells which spontaneously can beat are skeletal muscle cells differentiated from muscle progenitor cells which also belong to the mesodermal lineage.

Reviewer #2 (Remarks to the Author):

SUMMARY

This manuscript describes efforts to develop assisted reproductive technologies (ART) to rescue the northern white rhinoceros (NWR), an almost extinct species. Oocytes were harvested from southern white rhinos (SWR) by transrectal ovum pickup (OPU), matured, and fertilized by intracytoplasmic sperm injection (ICSI). The resultant embryos developed to the blastocyst stage. Next, hybrid rhino embryos were generated in vitro using NWR and SWR gametes. Cell lines were also established from blastocyst generated with SWR oocytes and NWR sperm that exhibited typical embryonic stem (ES) cell features. Some purebred and hybrid blastocysts were frozen for later transfer.

OVERALL COMMENTS

This manuscript details a tremendous effort to develop ART approaches with the goal of saving an endangered species. In general, the manuscript is well written with a logical set of experiments and quality methods and results. The major claims, experimental outcomes and their findings are justified by the results, novel and appropriately discussed. However, there is some question about whether the findings are of wide interest.

R# Thank you for the positive appreciation of our work. We provide a link to a recent news on newspaper and the interest of the media (not necessarily restricted to scientists) that suggests there is indeed wide interest in the subject. We suggest to have a look at the websites on March 20th, the day

the last NWR male was euthanized of all major daily newspapers. For example:
<https://www.theguardian.com/commentisfree/2018/mar/20/sudan-northern-white-rhino-dead-species-endangered-species-conservationists>

Major Comments:

(1) Do the authors have any experimental proof that the generated ES cell lines will contribute to an embryo when injected into a blastocyst? This experiment has to be done to establish that the derived ES cells will contribute to the germline and useful to regenerate individuals.

R# This is the ultimate experiment. However, we have to be realistic. If we exclude the mouse, germ line chimeras transmission have only been partially successful in rats but never proved on other mammalian species. Based on this argument, none of the human iPSC or ESCs should be used for anything because nobody has yet demonstrated this. Human ES cells have been validated and are accepted without the proof of making germ line transmission, for obvious reasons.

Reviewer #3 (Remarks to the Author):

This manuscript documents a huge effort in advancing assisted reproduction technology in rhinoceros. It clearly represents an exercise in teamwork by a number of knowledgeable experts. This kind of research represents the only hope of keeping some species of mammals from extinction. While success rates of some of the steps were on the low side, photomicrographs of the inner cell masses of several blastocysts document excellent morphology. The health of these cells also is documented by the cell lines produced, lines that have many of the properties of bona fide embryonic stem cells. Whether these cell lines will lead to germ line transmission remains to be seen, but this appears to be likely, one way or another. That these cell lines are normal diploid in chromosome composition, with the sperm and oocyte contributing is well documented. Presumably the authors are thinking about how to get viable pregnancies from this material.

R# We thank this referee for their appreciation of the efforts put into this work and deep understanding of the subject

I have embarrassingly little to add to improve the manuscript. I do suggest the following rewording:

L 51- embryos

R# changed

L 61- ...challenges; there are only...

R# changed

L 62- ...bigger than the horse and

R# changed

L 63- deleted duplicated this species

R# changed

L 68- sources of stem cells

R# changed

L 107- cleavage,
R# changed

Reviewer #4 (Remarks to the Author):

This is certainly an interesting and timely report describing approaches to try to rescue an endangered species for which few individuals remain, namely the Northern White Rhino. Fortunately, in this case the closely related Southern White Rhino is currently much more numerous and available to help in the project. In the paper the authors report on several approaches including in vitro fertilization using intracytoplasmic sperm injection (ICSI), and also the derivation of embryonic stem (ES) cells, which could potentially be used in the future to produce artificial gametes. Such ES cells could also provide a gold standard against which to characterize iPS cells if they are derived in the future.

R# iPS have already been reported see ref 22 (revised manuscript)

Clearly the authors are working in a ‘non-ideal’ situation with only two female and one Northern White Rhino still surviving. However, they have successfully developed methods for producing rhino embryos to a blastocyst stage for the Southern White Rhino, and hybrid embryos from Southern White Rhino eggs fertilized with Northern White Rhino sperm. Unfortunately the quality of available Northern White Rhino eggs precluded obtaining embryos from them. In addition they were able to derive putative ES cell lines from the Southern White Rhino embryos.

These results seem to me to be a laudable first step. However, the authors might better discuss how they will be used going forward. Clearly hybrid embryos, if allowed to develop to adult rhinos, will not replicate the Northern White Rhino: what strategies do the authors envisage to recreate that species. Likewise how do they envisage using ES cells or iPS cells: at the moment iPS cells would be the only way to capture the whole Northern White Rhino genome, but they have yet to be made, and the ability to generate functional gametes also lies in the future. Nevertheless, the newly created rhino ES cells will provide an important reference point for future work with iPS cells.

R# We repeat the answer to reviewer 1. We detailed these future steps in Ref 9, a white paper on the strategy we are developing. There are somatic cells of at least a dozen NWR individuals in cell banks. iPS cells have been already developed in rhinoceros (Ref 22 in revised paper) and mice have now been generated from iPS derived artificial gametes (Ref 10). In the latter reference it has been clearly shown that the development of a successful germ cell differentiation protocol takes several years and that the chances of getting this cell type dramatically increases by using ESC instead of iPSC. This is the long term goal and what we are presenting here are some cornerstones of what is required. We agree it will be a long process, and might not be achievable, but it is an excellent start.

However, given the importance of these new lines, in my view, their current characterization is insufficient:

In Figure 3, pictures of these new ES cells are provided. However, as seems common in the literature these days, the magnification is far too low to be able to make out the cells. ES cells have a fairly typical morphology – but that require high magnification pictures to see.

R# We provide some of the same pictures taken with a 40X objective as a separate file that for reasons of editorial policy can not be added to the paper (10 display item max.)

In the same figure, immunostaining for Oct4, Nanog, Sox2 and the surface antigen SSEA3 is shown. However, first, no negative control staining is shown. Second, there is nothing to confirm the

specificity of the antibodies: these are rabbit polyclonal antibodies, so the absence of staining by irrelevant heterotypic antibodies must be shown – at least in the case of the Abcam anti-Oct4 antibody, the artificial peptide used as an immunogen is available as a blocking peptide to test the specificity of staining.

R# We repeat the answer to reviewer 1. The ES cell staining was performed on ES cells growing on mouse feeder cells. While ES cells are strongly positive the feeder cells are clearly negative for OCT4, SOX2 and NANOG demonstrating the specificity of the antibody for undifferentiated nuclei. The same antibodies were used to stain the neural precursors derived from ES cells and in that case the staining was negative demonstrating that the secondary antibody per se cannot stain rhino cells. Moreover the nuclear staining of OCT4, NANOG and SOX2 is very typical of ES cells because nuclei are intensively stained while nucleoli are negative. In all instances the underneath mouse feeders are negative. The antibody staining was performed in Cremona with the antibody presented in the SI but also duplicated in Berlin with a completely different source of antibody listed here (except for the SSEAs that was the same) with the same results.

Oct-3/4 (H-134), rabbit polyclonal, Santa Cruz: sc-9081

- anti-OCT4, rabbit polyclonal, Life tech. Staining Kit; A24867
- Nanog, rabbit polyclonal, Thermo Fisher: PA1-097
- Nanog, goat polyclonal, R&D Systems: AF1997
- SOX2 (NH2 terminus), rabbit polyclonal, BioLegend: 630802
- anti-SOX2, rat monoclonal, Life tech. Staining Kit; Bestellnummer: A24759
- SSEA3 MC631; rat monoclonal, Thermo Fisher: MA1-020x
- SSEA4 MC-813-70; mouse monoclonal, Thermo Fisher: MA1-021x

We have provided as an additional file (because we have already reached the limit of 10 display items indicated on the instruction to authors) the staining only with the secondary antibody in the absence of the primary to prove the specificity of the binding as requested as well as fibroblasts staining and a higher magnification of the undifferentiated rESC.

How do we know that these antibodies made to mouse proteins will cross react with the rhino equivalents? Further, as we know little about expression of these markers in rhino embryos, there is a leap of faith in assuming that their expression patterns in rhinos will match that in other species – a reasonable expectation but it should be demonstrated.

R#. There is abundant literature reporting the staining of livestock embryos including the horse⁸ where commercially available antibodies for mouse and or human react with specificity to undifferentiated cell into the embryos. These pluripotency markers are highly conserved across mammalian species from humans through horses to marsupials. There is no reason to suspect that rhinos are any different. Also we refer the reviewer to Ref 22.

In fact in the case of SSEA3, it is a good marker for human ES cells but it is not expressed by mouse ES cells: we clearly do not know how it will behave in rhinos. I would suggest that a better approach would be to collect data on expression of a wider range of genes, perhaps by RNAseq, which would then provide a better way to compare with ES cells of other species.

R# rESC are similar to human, not mouse, they are FGF dependent and behave similarly in culture and this is confirmed by SSEA3 staining as we also see in bovine ES cells. In an ideal situation RNAseq would be the final word however again we must be realistic, we know little about the genome annotation of the rhino and it will require a lot of work and a lot of time that goes beyond the scope of this paper.

More importantly than characterizing marker expression is a functional demonstration of pluripotency.

The generally accepted way to characterize a pluripotent stem cell is to show differentiation into derivatives of all three germ layers. In the mouse this is generally done by making chimeric embryos. This would clearly be a challenge in the rhino but the approach of making xenograft teratomas in immunodeficient mice is generally used for human ES cells and could easily be tried with the rhino ES cells. As it is, the authors have just done limited in vitro differentiation, assessed by immunostaining for just one marker for each germ layer (with the same caveats as above regarding lack of negative controls and evidence of antibody specificity).

Overall, in my view, the derivation of the rhino ES cells is an important result, but these lines deserve a more detailed characterization as summarized above.

R# We agree with the referee that more work can be done if material was available. However we respectfully disagree with the lack of specificity for the antibody staining for the reasons mentioned above. We have demonstrated differentiation of these cells with the beating cardiomyocytes video that demonstrates beyond any reasonable doubt of any staining or any specificity that these cells are differentiated and beating cells. We have now included as an additional file for controls, staining with the secondary antibodies without the primary antibodies that are all negative, and the staining of rhino fibroblasts that are also negative and note that the mouse feeders are also negative, It has to be highlighted here that we were using protocols meant and developed to differentiate the targeted cells into the lineage of interest, and that we got the expected cell type by using these protocols. Moreover in a recent publication it has been shown that the significance of the teratoma assay might be questionable and strongly depends on number of cells injected and also the area in which the cells were injected⁹. Because of that there is currently a debate among the stem cell community about the necessity of this assay¹⁰. That's why more recently researchers started to use alternative direct approaches, avoiding the need to unnecessarily sacrifice mice¹¹. Because of the reasons mentioned above these direct approaches are becoming more and more standard best practice recognized by stem cell banks and stem cell registries (<https://hpscereg.eu/>).

However the scope of our work was, besides producing good quality embryos in a reproducible manner, to develop embryo transfer technique to replace the embryos in the uterus of a SWR female, and demonstrate their potential viability by establishing undifferentiated pluripotent cells with bona fide markers established in the literature. We clearly show without any reasonable doubt the characteristics of our cells match the definition of ESCs described for other mammals with classical well established markers with high specificity. Because our long term project and efforts are devoted not to induce terminally differentiated cells but to coax these cells into primordial germ cells, we decided not to undertake this route but to concentrate towards PGC differentiation.

- 1 Groves, C. P., Fernando, P. & Robovsky, J. The sixth rhino: a taxonomic re-assessment of the critically endangered northern white rhinoceros. *PLoS one* **5**, e9703, doi:10.1371/journal.pone.0009703 (2010).
- 2 Lee, J.-W., Tian, X. C. & Yang, X. Optimization of parthenogenetic activation protocol in porcine. *Molecular Reproduction and Development* **68**, 51-57, doi:10.1002/mrd.20043 (2004).
- 3 Zhang, J. *et al.* Electrical activation and in vitro development of human oocytes that fail to fertilize after intracytoplasmic sperm injection. *Fertility and Sterility* **72**, 509-512, doi:[https://doi.org/10.1016/S0015-0282\(99\)00264-2](https://doi.org/10.1016/S0015-0282(99)00264-2) (1999).

- 4 Vanden Meerschaut, F. *et al.* Neonatal and neurodevelopmental outcome of children aged 3–
10years born following assisted oocyte activation. *Reproductive BioMedicine Online* **28**, 54-63,
doi:<https://doi.org/10.1016/j.rbmo.2013.07.013> (2014).
- 5 Vanden Meerschaut, F., Nikiforaki, D., Heindryckx, B. & De Sutter, P. Assisted oocyte
activation following ICSI fertilization failure. *Reproductive BioMedicine Online* **28**, 560-571,
doi:<https://doi.org/10.1016/j.rbmo.2014.01.008> (2014).
- 6 Blondin, P., Bousquet, D., Twagiramungu, H., Barnes, F. & Sirard, M. A. Manipulation of
follicular development to produce developmentally competent bovine oocytes. *Biology of
reproduction* **66**, 38-43 (2002).
- 7 Bongso, T. A., Hilmi, M. & Basrur, P. K. Testicular cells in hybrid water buffaloes (*Bubalus
bubalis*). *Research in veterinary science* **35**, 253-258 (1983).
- 8 Choi, Y. H. *et al.* Cell lineage allocation in equine blastocysts produced in vitro under varying
glucose concentrations. *Reproduction (Cambridge, England)* **150**, 31-41, doi:10.1530/rep-14-
0662 (2015).
- 9 Bouma, M. J. *et al.* Differentiation-Defective Human Induced Pluripotent Stem Cells Reveal
Strengths and Limitations of the Teratoma Assay and In Vitro Pluripotency Assays. *Stem cell
reports* **8**, 1340-1353, doi:10.1016/j.stemcr.2017.03.009 (2017).
- 10 Buta, C. *et al.* Reconsidering pluripotency tests: do we still need teratoma assays? *Stem cell
research* **11**, 552-562, doi:10.1016/j.scr.2013.03.001 (2013).
- 11 Tsankov, A. M. *et al.* A qPCR ScoreCard quantifies the differentiation potential of human
pluripotent stem cells. *Nature biotechnology* **33**, 1182-1192, doi:10.1038/nbt.3387 (2015).

Reviewers' comments:

Reviewer #1 (Remarks to the Author):

Overall comments:

The authors have done a commendable job addressing most of the comments/questions in my original review. The addition of more details, negative control data and figures to the text and supplementary information make the results much more convincing and, in general, I no longer have questions regarding the validity of the work reported herein. It still does give me pause that the development of the hybrid embryos was slightly retarded and that ICM cell line could not be produced from them, but the authors did add a very plausible explanation for the latter since the hybrid blastocysts were plated out earlier than the SWR blastocysts despite developing more slowly. I guess we will eventually find out their ultimate viability/vigor when they are transferred into surrogate females.

In my opinion, there are still a few statements that represent exaggerated or inaccurate claims regarding the results and progress demonstrated by the data presented herein. I do understand and support the authors' desires to give people hope, I just think as scientists we need to remain conservative and restrict conclusions to "evidence based observations" (to use the authors' own words). However, it also occurred to me that some of our differences in opinions could partly be due to semantics, so I have taken the liberty to make suggestions regarding the revision of the statements of concern (see specifics below). I hope the authors do not take offense. I am trying to reach a middle ground regarding our differing views on these few remaining issues.

Regarding the species/sub-species issue – I am well aware of Colin Groves paper and argument/evidence that the Northern white rhino should be a separate species. It is relatively common for different geneticists to have different views regarding the taxonomic designation of wildlife species and sub-species, and those views change over time with the advancement of technology and the science itself. That is why IUCN designation is used as the gold standard. The IUCN listens to all views, looks at all data and considers all information from those with varied opinions and then does its best to come to the right conclusion. We all know these things can and do change over time, but currently, the IUCN has the Northern white rhino listed as a sub-species. Furthermore, if the two were different species, I think the authors' strategy of hybridization to save the species would attract significantly more criticism from the conservation community as there is much more resistance to mixing species than sub-species. Therefore, I do not think it is in the authors' best interest to argue for the separate species delineation.

Specifics:

1) Title – related to the above, I suggest the authors change the title to read: "Embryos and stem cells from the white rhinoceros" or if they prefer "Embryos and stem cells from the imperiled white rhinoceros" since one can certainly argue that all white rhinos are imperiled given the poaching crisis, and the word "imperiled" is not a part of the IUCN official endangered classification so it cannot be considered inaccurate.

2) Line 32 – you may want to update this to just two females surviving.

- 3) Line 33-34 – suggest changing to: "...a common ancestor. Therefore, assisted reproduction techniques (ART) developed in equines can potentially be translated to rhinoceros species, including the southern white rhinoceros..."
 - 4) Line 44-46 – suggest changing to: "Our results indicate that ART could be a viable strategy to rescue genes from the iconic, almost extinct, Northern white rhinoceros and may also have broader impact if applied with similar success to other endangered large mammalian species."
 - 5) Line 52 – suggest changing to: "...suggested that ART might offer an option for rescuing genes from this nearly extinct rhinoceros sub-species." Or, if the authors prefer and word count allows, "...suggested that ART might offer an option for rescuing genes from the NWR, an essential first step in saving this nearly extinct rhinoceros sub-species."
 - 6) Line 56 – suggest changing "ensure sustainable" to "maximize"
 - 7) Line 88 – omit redundant clause "soon after sperm injection"
 - 8) Line 94 – to what does "they" refer to? mNWR3 "samples"?
 - 9) Line 149-150 – suggest changing to: "...attempted in this threatened species. Our results suggest that these methods could play a valuable role in the effort to save rhinoceros populations on the brink of extinction."
 - 10) Line 151-152 – suggest changing to "...the potential of rescuing genes of the NWR sub-species through intra-specific hybrid breeding."
 - 11) Line 154 – suggest changing "foster" to "surrogate" for accuracy.
- Table S3 – there is a typo in "cleavage" in the footnote.

Reviewer #2 (Remarks to the Author):

The revised manuscript has addressed many of the comments from the initial review, however lingering issues remain with the precise novelty of the claims and whether it represents a true advance in the field. Of paramount concern is that the generated embryonic stem (ES) cells may not truly be ES cells, and no evidence is provided to support this claim.

Reviewer #4 (Remarks to the Author):

As I implied in my review of the manuscript that the authors originally submitted, I found their approach both interesting and important. Nevertheless, if they are going to report the derivation and characterization of embryonic stem (ES) cells from the Northern White Rhino, which would become an important tool for future research and for achieving the aims to which the authors are working, then they must provide appropriate data to support their claims.

Regarding the points I made, the authors have now provided higher magnification photomicrographs of their NWR-derived cells which I agree do now allow the reader to see a morphology typical of ES cells from other species. However, in my view the authors have not adequately addressed my concerns about the immunostaining for OCT4, NANOG and SOX2. That their antibodies, which are mostly polyclonal, do not stain mouse embryonic

fibroblasts, nor indeed neural neural precursors derived from their NWR ES cells, does not prove that they are recognizing OCT4, NANOG and SOX2, respectively in their ES cells. Particularly with polyclonal sera there are ample opportunities for heterophile antibodies causing confusion. The argument that the antibodies will cross react with the rhino proteins because they cross react with other livestock species is also weak.

More important than expression of markers, pluripotent stem cells, whether ES or iPS cells, are defined as pluripotent because of their ability to differentiate into derivatives of all three germ layers. In my view confirming pluripotency by staining for one marker of each germ layer is inadequate, even if some beating cells are also formed. Despite the comments made by the authors, the teratoma assay is widely used and provides a well-established and robust method for showing multi-lineage differentiation potential. If they do not like this approach then others have used other in vitro assays such as differentiation in embryoid bodies in which a wide variety of cells may be produced and these can be assessed by gene expression patterns.

No doubt the results reported by the authors are, in aggregate, consistent with their conclusion that the cells they have isolated are ES cells. However, it seems to me that these cells will potentially be an important reference point for people working with rhino ES and iPS cells in the future – so why not do the experiments properly so the results are conclusive.

Reviewers' comments:

We thank all the reviewers for their time dedicated to us, constructive comments and helpful suggestions that have improved the manuscript.

Reviewer #1 (Remarks to the Author):

Overall comments:

The authors have done a commendable job addressing most of the comments/questions in my original review. The addition of more details, negative control data and figures to the text and supplementary information make the results much more convincing and, in general, I no longer have questions regarding the validity of the work reported herein. It still does give me pause that the development of the hybrid embryos was slightly retarded and that ICM cell line could not be produced from them, but the authors did add a very plausible explanation for the latter since the hybrid blastocysts were plated out earlier than the SWR blastocysts despite developing more slowly. I guess we will eventually find out their ultimate viability/vigor when they are transferred into surrogate females.

In my opinion, there are still a few statements that represent exaggerated or inaccurate claims regarding the results and progress demonstrated by the data presented herein. I do understand and support the authors' desires to give people hope, I just think as scientists we need to remain conservative and restrict conclusions to "evidence based observations" (to use the authors' own words). However, it also occurred to me that some of our differences in opinions could partly be due to semantics, so I have taken the liberty to make suggestions regarding the revision of the statements of concern (see specifics below). I hope the authors do not take offense. I am trying to reach a middle ground regarding our differing views on these few remaining issues.

R# we appreciate the fair-play of rev #1. We are not offended and we are open to his/her suggestions and have taken these into consideration and amended the text.

Regarding the species/sub-species issue – I am well aware of Colin Groves paper and argument/evidence that the Northern white rhino should be a separate species. It is relatively common for different geneticists to have different views regarding the taxonomic designation of wildlife species and sub-species, and those views change over time with the advancement of technology and the science itself. That is why IUCN designation is used as the gold standard. The IUCN listens to all views, looks at all data and considers all information from those with varied opinions and then does its best to come to the right conclusion. We all know these things can and do change over time, but currently, the IUCN has the Northern white rhino listed as a sub-species. Furthermore, if the two were different species, I think the authors' strategy of hybridization to save the species would attract significantly more criticism from the conservation community as there is much more resistance to mixing species than sub-species. Therefore, I do not think it is in the authors' best interest to argue for the separate species delineation.

R# We agree. We did list the NWR as sub species in the revised version.

Specifics:

1) Title – related to the above, I suggest the authors change the title to read: "Embryos and stem cells from the white rhinoceros" or if they prefer "Embryos and stem cells from the imperiled white rhinoceros" since one can certainly argue that all white rhinos are imperiled given the poaching crisis, and the word "imperiled" is not a part of the IUCN official endangered classification so it cannot be considered inaccurate.

R# We are happy to add the “white” - it will be even more direct and inclusive. Thank you for the suggestion. For accuracy I also added “embryonic” to the stem cells

2) Line 32 – you may want to update this to just two females surviving.

3) Line 33-34 – suggest changing to: “...a common ancestor. Therefore, assisted reproduction techniques (ART) developed in equines can potentially be translated to rhinoceros species, including the southern white rhinoceros....”

4) Line 44-46 – suggest changing to: “Our results indicate that ART could be a viable strategy to rescue genes from the iconic, almost extinct, Northern white rhinoceros and may also have broader impact if applied with similar success to other endangered large mammalian species.”

5) Line 52 – suggest changing to: “...suggested that ART might offer an option for rescuing genes from this nearly extinct rhinoceros sub-species.” Or, if the authors prefer and word count allows, “...suggested that ART might offer an option for rescuing genes from the NWR, an essential first step in saving this nearly extinct rhinoceros sub-species.”

6) Line 56 – suggest changing “ensure sustainable” to “maximize”

7) Line 88 – omit redundant clause “soon after sperm injection”

8) Line 94 – to what does “they” refer to? mNWR3 “samples”?

9) Line 149-150 – suggest changing to: “...attempted in this threatened species. Our results suggest that these methods could play a valuable role in the effort to save rhinoceros populations on the brink of extinction.”

10) Line 151-152 – suggest changing to “...the potential of rescuing genes of the NWR sub-species through intra-specific hybrid breeding.”

11) Line 154 – suggest changing “foster” to “surrogate” for accuracy.

Table S3 – there is a typo in “cleavage” in the footnote.

R# all the suggested changes have been introduced in the paper

Reviewer #2 (Remarks to the Author):

The revised manuscript has addressed many of the comments from the initial review, however lingering issues remain with the precise novelty of the claims and whether it represents a true advance in the field. Of paramount concern is that the generated embryonic stem (ES) cells may not truly be ES cells, and no evidence is provided to support this claim.

R# We understood from the previous comments that rev#2 was satisfied with the work, therefore in our reply to rev #2 we only stated that germ line chimeras are a not reasonable expectation when working with ES cells of an endangered species such as the rhino when we have just produced the few first ever seen pre-implantation embryos. In any case this request was overruled by the Editor as out of the scope of the paper. We and the other 3 reviewers think that the findings reported in this paper are novel and advancing the field. Having ES cells from these animals clearly increases our chance to derive oocytes and sperm later on since it has been shown that the efficacy of germ cell differentiation is higher in ESCs than in iPSCs (ref 10). Therefore, we hope that these cells will help us to develop a reliable germ cell differentiation protocol which can be adopted subsequently to iPSCs.

Additionally, we have toned down the claims of the success in saving the subspecies or extending it to other large mammals. We have modified the title, see also line 44-46, line 52, line 149-152 for the changes.

We summarize the experimental work done to characterise the ES cells and the additional experiments done after the first review to support our data as requested by the rev #1 and #4 as it is now the subject of the discussion that has been extensively addressed by rev#1 and 4.

One of the 2 ES cell lines has been kept in culture since its derivation (March 2017) and aliquots repeatedly frozen and thawed of both cell lines. So far, we didn't see any differences in terms of proliferation ability or differentiation potential. In order to validate the pluripotency of our cells we used the canonical markers OCT4, Nanog, Sox2, SSEA3 in ICC both with polyclonals (OCT4, Nanog, Sox2, see fig 3e, 3i, 3o) and with monoclonals (OCT4, Sox2, SSEA3 see fig A11a, b and fig 3n), confirming the specificity of our staining. Negative controls without primary antibody were performed and also ICC with the same antibodies was performed on rhino fibroblasts with negative results (see fig. 2 CTR2). To further confirm the specificity, we also performed RT-PCR that show an upregulation of the 3 pluripotency markers compared to fibroblasts (fig A12). In order to show the ability of the cells to differentiate into all 3 germ layers, the other hallmark of pluripotency, we either took advantage of embryoid bodies or direct differentiation based technologies. For the ectodermal derivatives we have shown the formation of neural rosettes from embryoid bodies (fig A13) and after differentiation the presence of 4 markers: P75; Tuj1 and TH (see fig 3s, 3t, 3u) and Peripherin (a monoclonal, fig A14). We have done 3 markers for both endoderm GATA4, GATA6 and Sox17 (see fig 3v, 3z, 3x) and for mesoderm, Myosin, Nkx2.5 and Tnnt2 (see fig 3y and 3w). Therefore we have fulfilled many, if not all necessary criteria to confirm the true nature of our rhino ES cells based on the recommendation formulated by NIH (<https://stemcells.nih.gov/info/basics/3.htm>)

1. Proliferate indefinitely = self renewal
2. Recover after freezing thawing cycles
3. Normal karyotype
4. Positive for pluripotency markers
5. Differentiate into the 3 germ layers

We acknowledge that there are many more assays, markers and information that can be acquired to fully characterize rES cells we generated and indeed it might be life long work and the subject of many other papers but “at this moment in time” what we have provided so far is rather compelling.

Reviewer #4 (Remarks to the Author)

As I implied in my review of the manuscript that the authors originally submitted, I found their approach both interesting and important. Nevertheless, if they are going to report the derivation and characterization of embryonic stem (ES) cells from the Northern White Rhino,

R# For clarity, please note that we have derived Southern White Rhino (SWR) ES cells (not NWR) as stated in the introductory paragraph and later in the manuscript

which would become an important tool for future research and for achieving the aims to which the authors are working, then they must provide appropriate data to support their claims. Regarding the points I made, the authors have now provided higher magnification photomicrographs of their NWR-derived cells which I agree do now allow the reader to see a morphology typical of ES cells from other species.

However, in my view the authors have not adequately addressed my concerns about the immunostaining for OCT4, NANOG and SOX2. That their antibodies, which are mostly polyclonal, **R# As reference, I point you to ref 22, a publication in Nature Methods demonstrating the use of both polyclonal and monoclonal antibodies to characterise rhino iPS cells. Moreover we noticed, comparing the quality of our immunocytochemistry (it is the protein that matters) to that of the Nature Methods paper, that our Figures of the staining for OCT4 and Nanog at high magnification (even when enlarged to 400%) are of textbook quality imaging of the nucleus (intense staining and negative cytoplasm) and**

nucleoli (not stained for Oct4-Nanog as expected).

do not stain mouse embryonic fibroblasts, nor indeed neural precursors derived from their NWR ES cells, does not prove that they are recognizing OCT4, NANOG and SOX2, respectively in their ES cells.

R# Our antibodies do not stain rhino fibroblasts and have provided this data in the last reply letter (fig 2 CTR2)

Particularly with polyclonal sera there are ample opportunities for heterophile antibodies causing confusion. The argument that the antibodies will cross react with the rhino proteins because they cross react with other livestock species is also weak.

R# This argument could be discussed for a long time with little resolution. We all know that monoclonal or polyclonal antibodies each have their pros and cons. When working with species other than the mouse or humans, polyclonals have more opportunities to capture the protein of interest for most antigens. If the monoclonal works, this is even better but that is not always the case. We have already used 2 monoclonal antibodies: SSEA3 (see fig 3n) and we indicated in the rebuttal letter that our colleagues in Berlin also used SOX2 (attached now is also the ICC picture, fig AI1e, that was not used for fig3). For SSEA there are 2 monoclonals, 1 and 3, one recognizes the mouse antigen and the other the human (and we used it for bovine cells and it works). SSEA stands for Surface Specific Embryonic Antigen. If you look at fig 3n, it is beyond any reasonable doubt that it is on the surface of the cells and the nuclei are not stained. Moreover, to respond in full about the specificity issue we have screened a series of monoclonals (as required by rev #4) and we provide the data in the file “additional information and experimental data” as fig AI1. While the Nanog monoclonal did not work, the OCT4 and Sox2 did. So in total we have 3 monoclonals for 3 pluripotency markers that have worked, confirming our previous findings with polyclonals. In addition to ICC we performed RT-PCR for the 3 pluripotency markers (OCT4, Nanog and SOX2) and the results are shown in fig AI2, where it is clearly shown a strong upregulation of these markers compared to fibroblasts.

More important than expression of markers, pluripotent stem cells, whether ES or iPS cells, are defined as pluripotent because of their ability to differentiate into derivatives of all three germ layers. In my view confirming pluripotency by staining for one marker of each germ layer is inadequate, even if some beating cells are also formed.

Despite the comments made by the authors, the teratoma assay is widely used and provides a well-established and robust method for showing multi-lineage differentiation potential. If they do not like this approach then others have used other in vitro assays such as differentiation in embryoid bodies in which a wide variety of cells may be produced and these can be assessed by gene expression patterns.

R# We ruled out doing the teratomas for the reasons discussed in our last letter. I understand that reviewer #4 accepts this now. If it were to be required, it will take months, millions of cells and dozens of mice to obtain at the end a random differentiation. I hope you agree that this is an unnecessary and costly delay.

We contend that differentiated markers are more important than undifferentiated one. You can transdifferentiate a cell without bring it to a pluripotent state, so we believe that the canonical undifferentiated markers we analyse are the “condicio sine qua non” you can claim you have a ES cell line. To generate neural precursors Embryoid Bodies were indeed made (see Supplementary Information file point n 8) and then coaxed to neural differentiation. We now provide in fig AI3 the intermediate stage of neural rosette formation as a proof that we obtained embryoid bodies, then neural

rosette (ref 21) and finally neural derivatives. Endodermal and mesodermal differentiation was achieved by direct induction. Direct differentiation is the standard approach if you want to generate cells to be used for any purpose including primordial germ cell differentiation, which is our primary long term objective. We however point out that we have provided **3** stainings for each lineage, not **one** as indicated by reviewer 4: for neural/ectoderm tissue we used P75; Tuj1 and TH (see fig 3s, 3t, 3u); for endoderm GATA4, GATA6 and Sox17 (see fig 3v, 3z, 3x); and for mesoderm, Myosin, Nkx2.5 and Tnnt2 (see fig 3y and 3w), and we have done more (BLBP, AFP, FoxA2) that for reason of space constrain were not included. The statement “**In my view confirming pluripotency by staining for one marker of each germ layer is inadequate, even if some beating cells are also formed**“ is not correct. Amongst the monoclonals that we screened we found Peripherin antibody working and the data are shown in fig AI4

□ No doubt the results reported by the authors are, in aggregate, consistent with their conclusion that the cells they have isolated are ES cells. However, it seems to me that these cells will potentially be an important reference point for people working with rhino ES and iPS cells in the future – so why not do the experiments properly so the results are conclusive. □

R# “**the results reported by the authors are, in aggregate, consistent with their conclusion that the cells they have isolated are ES cells**” we are pleased of this acknowledgment.

However we do not agree with this statement “**why not do the experiments properly** “. We have provided data using monoclonal antibodies and now provide additional experimental evidence with other antibodies, we have done the RT-PCR for the pluripotency genes and clarified that we analysed **3** markers for each lineage (now **4** for the ectodermal one with the addition of perpherin), not one as the reviewer contends. The characterisation of the differentiation potential is satisfactory when coupled with the typical ES morphology, culture behaviour (the cells have been kept in culture for almost a year) and that have a normal diploid karyotype, survive freezing and thawing, etc..

In our point of view, the directed differentiation approaches we used to derive the specialized cells is more valuable than taking advantage of random differentiation processes which can not be precisely controlled as the direct methods. Therefore, we invested quite some time to find the right conditions to direct the cells in the specific lineages we were interested to provide essential characterization data of our ESCs. These cells will be the foundation of our follow up project in which we want to compare NWR iPS and SWR ES cells on a molecular level to elucidate a proper directed controlled way to differentiate these cells into primordial germ cells and later oocytes. Additionally, the significance of the teratoma assay has been discussed and compared to alternative approaches like embryoid body formation and directed differentiation approaches. Directed differentiation approaches are therefore widely accepted as an alternative approach to the random differentiation methods for confirmation of pluripotency¹.

The primary goal of this study was to derive viable embryos from an endangered species with the ultimate goal to implant them into a surrogate mother. Establishing the ESCs was one way of confirming the quality of the embryos.

We did not invest a lot of time and resources in the differentiation as it was not the primary scope of the paper but we do provide the essential state-of-the-art characterization. We wanted to show that the embryos were viable through ES cell derivation because we are now developing the embryo transfer technique (another big challenge) and also we want to obtain artificial gametes. Therefore the generation of pluripotent stem cells, characterised as above by typical ES morphology and high quality ICC images, is an indirect assessment of that and the gold standard for comparison with iPS lines already available or that will be made in the future. Our project is focussed to direct ES cells towards

primordial germ cell differentiation. This is why we were not particularly interested in studying in detail random terminal differentiation. One of us is the author of ref 10 that is why we concentrate on direct differentiation with the focus at PGC differentiation. We think that this paper provides a clear example of a multidisciplinary approach using advanced techniques: basic reproduction of an endangered species and customized clinical gynaecology, assisted reproduction technologies and embryonic stem cell biology, to tackle a complex biological issue and sets an example of how modern biotechnologies can combine and contribute to conservation strategies.

- 1 Buta, C. *et al.* Reconsidering pluripotency tests: do we still need teratoma assays? *Stem cell research* **11**, 552-562, doi:10.1016/j.scr.2013.03.001 (2013).

REVIEWERS' COMMENTS:

Reviewer #4 (Remarks to the Author):

The authors have provided some additional data about differentiation potential that help to support their contention that the cell lines they have derived are embryonic stem cells, and I think this is adequate within the context of their paper, although I still contend that a more detailed characterization using either teratoma or EB differentiation assayed for expression of multiple genes as in the now well established score card assays would have been better, and to my mind feasible. Staining with additional monoclonal antibodies to OCT4 and SOX2 and PCR data are provided in a file of additional information, but I do not see a call out to these data in the manuscript nor any information in the supplementary materials and methods. This omission should be corrected.

REVIEWERS' COMMENTS:

Reviewer #4 (Remarks to the Author):

The authors have provided some additional data about differentiation potential that help to support their contention that the cell lines they have derived are embryonic stem cells, and I think this is adequate within the context of their paper, although I still contend that a more detailed characterization using either teratoma or EB differentiation assayed for expression of multiple genes as in the now well established score card assays would have been better, and to my mind feasible. Staining with additional monoclonal antibodies to OCT4 and SOX2 and PCR data are provided in a file of additional information, but I do not see a call out to these data in the manuscript nor any information in the supplementary materials and methods. This omission should be corrected.

R# we have put a call out of these data in the main manuscript and also put in fig 3 of the main manuscript the RT-PCR image.